# Output Alignment: A Top-down Approach to Length Generalization

## Abstract

Recently, large language models have exhibited impressive performance and surprising emergent properties. However, their abilities remain constrained by the preset context window of the Transformer architecture, and they continue to struggle with length generalization. In this work, we propose a new perspective on length generalization by focusing on the output distribution rather than the input, as most prior studies have done (e.g., through positional encodings or data structure). First, through case studies on simple synthetic tasks, we highlight the importance of **output alignment**—the consistency of output distributions across sequences of varying lengths. We then extend this observation to natural language tasks and introduce a metric named Long-Short Misalignment to quantify output alignment, finding a strong correlation between this metric and length generalization performance. Based on these insights, we propose a regularization loss during training that improves output alignment. Extensive experiments confirm the effectiveness of this approach. Overall, our work provides a novel perspective for understanding and enhancing length generalization in large language models.

## 1 Introduction

Large language models (LLMs) have demonstrated impressive abilities in various tasks such as natural language generation, reading comprehension, code synthesis, instruction-following, and commonsense reasoning (Radford et al., 2019; Brown et al., 2020; Chowdhery et al., 2023; Touvron et al., 2023). Their performance has consistently improved by scaling both model and dataset sizes (Kaplan et al., 2020). However, the ability to generalize from smaller training context sizes to larger ones, commonly known as length generalization, is a major challenge for Transformer-based language models (Anil et al., 2022). This issue persists even in larger Transformers (Liu et al., 2024). With larger context sizes, a model can benefit from more in-context learning examples, an increased number of reasoning steps, or longer text generation (Li et al., 2024; Huang & Chang, 2023). However, training a Transformer with a larger context size is often excessively slow and memory-intensive. Therefore, understanding the mechanism of length generalization and enhancing the length generalization ability of these models is urgently needed.

There exist two main approaches to understanding and improving length generalization. The first is to understand and design different positional encoding (PE) (Press et al., 2021; Su et al., 2024; Kazemnejad et al., 2023; Peng et al., 2024; Chen et al., 2024; Yang, 2023; Zhang et al., 2024b). PE plays a crucial role in the length generalization of Transformers, as the model must systematically encode tokens in all possible positions. By designing PE that reduces the gap between shorter training sequences and longer test sequences, length generalization can be improved to some extent (Kazemnejad et al., 2023). The second approach is to understand the fundamental mechanisms of Transformer models (Zhou et al., 2024; Lee et al., 2023; Veličković & Blundell, 2021; Nogueira et al., 2021; Deletang et al., 2022). This includes studying which algorithms Transformers can represent and how they can generalize better by designing more effective tasks. However, in this work, we find that the previous works ignore a component, the output space of the model, which we found to be quite crucial in the length generalization tasks.

We begin by investigating length generalization on synthetic tasks: predicting the mean value and the length of binary sequences. Both empirical and theoretical results reveal that while Transformers generalize well in the mean prediction task, they struggle in the length prediction task. The key dis-

tinction lies in the output distribution's support set: in the mean prediction task, it remains consistent as input sequences grow, whereas in the length prediction task, the support set shifts with increasing sequence lengths. We hypothesize that this **misalignment in the output distribution leads to poor length generalization** in the length prediction task. To verify this hypothesis, we propose a reparameterization technique named OutRep that explicitly aligns the output distribution across different sequence lengths in the length prediction task. Our empirical and theoretical analyses confirm that this approach significantly enhances length generalization which supports our hypothesis.

Building on this insight, we extend our findings to natural language tasks. Although the model output in natural language tasks is a vector (unlike the scalar output in synthetic tasks), which makes it difficult to directly apply the same analysis, similar output misalignment issues still arise. Specifically, for two sequences with the same ending but slightly different lengths, the models output is expected to remain consistent. However, we find that models with poor length generalization tend to produce divergent outputs when conditioned on these sequences. To explore the quantitative relationship between output alignment and length generalization, we introduce a metric—long-short misalignment—using symmetrical cross-entropy loss to measure the divergence between the outputs of such sequences. Both empirical and theoretical results demonstrate that this metric strongly correlates with a models long-context performance, making it a more reliable predictor of length generalization than traditional training loss. Consequently, we incorporate this metric as a regularization loss in training, and extensive experiments on both length generalization and long-context modeling show that our proposed training loss significantly improves performance.

The main contributions of this work are as follows:

- We identify the crucial role of output alignment in achieving length generalization. Both empirical and theoretical analyses show that misalignment in the output distribution across varying input sequence lengths leads to poor generalization performance.

- Building on this insight, we introduce a long-short misalignment metric to quantify output misalignment and demonstrate its strong correlation with long-context modeling performance both empirically and theoretically.

- To further improve generalization, we integrate this metric as a regularization loss into the training process. Extensive experiments validate the effectiveness of this approach in boosting performance.

## 2 RELATED WORK

**Length Generalization on Synthetic Tasks.** Our paper is related to the line of work that seeks to understand the capabilities and limitations of Transformer models when it comes to algorithmic reasoning (Veličković & Blundell, 2021). Specifically, we focus on simple tasks and study length generalization on the standard Transformer architecture with causal structure. Related to this, Lee et al. (2023) study how well transformers trained from scratch can learn simple arithmetic tasks, and find that no length generalization is observed. Nogueira et al. (2021) find that partial length generalization on addition is observed only when models reach 3B parameters and when the addition questions are presented in reverse order. Jelassi et al. (2023) study models trained on addition and find strong generalization performance when using a few examples of longer sequences. Zhou et al. (2024) proposes the RASP Generalization Conjecture that Transformers tend to learn a length-generalizing solution if there exists a short RASP-L program that works for all input lengths. Liu et al. (2023) discovers that the Transformer will learn shortcuts through the study of various synthetic tasks. Besides these explorations on specific tasks, some works study the impact of different positional encodings on the length generalization of math reasoning tasks. Alibi adds a linear bias on the attention score to achieve better length generalization performance (Press et al., 2021). Ontanon et al. (2022) studies different settings of positional encodings and identifies Transformer configurations that generalize compositionally significantly better in a diverse set of compositional tasks. Kazemnejad et al. (2023) systematically studies the role of no positional encoding (NoPE) in Transformer with causal structure. Ruoss et al. (2023) proposes a randomized positional encoding scheme that simulates the positions of longer sequences and randomly selects an ordered subset to fit the sequence's length.

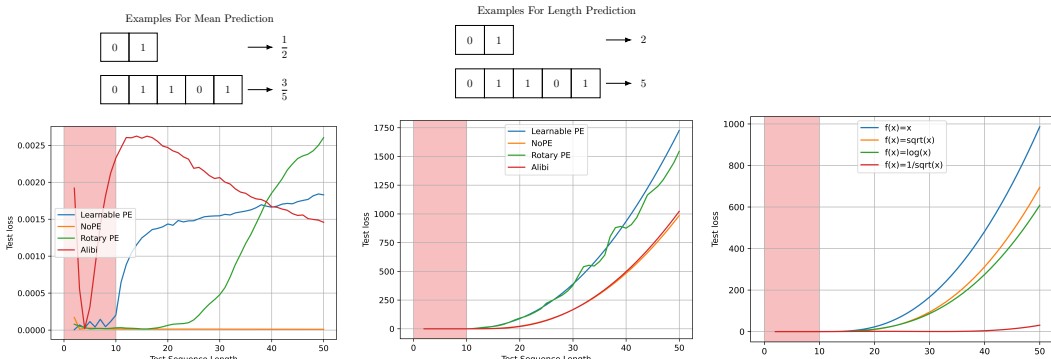

(a) Length generalization in the **mean** prediction task with different positional encodings

(b) Length generalization in the **length** prediction task with different positional encodings

(c) Length generalization in the **length** prediction task with different reparameterization function.

Figure 1: Comparison between the length generalization performance in the mean prediction task and the length prediction task. The training sequence length is uniformly selected from $[1, 10]$ (indicated by the light red area) while the test sequence lengths (on the x-axis) can reach a maximum of 50. In the length prediction task (b), the model struggles with length generalization, evidenced by an increasing test loss as sequence length increases. Conversely, the model demonstrates significantly better length generalization in the mean prediction task (a), maintaining consistent test loss regardless of the increasing sequence length using NoPE (indicated by the orange line). Figure (c) shows the length generalization performance in the length prediction task using different reparameterization functions $f(x)$. All three reparameterized targets improve generalization compared to the origin (blue) target. Among them, $f(x) = 1/\sqrt{x}$ (red) performs exceptionally well, significantly reducing test loss for longer input sequences.

**Long-context Modeling on Natural Language Tasks.** A series of works (Sun et al., 2023; Chi et al., 2022; 2023; Zhang et al., 2024b; Chen et al., 2024; Peng et al., 2024; Yang, 2023; Chen et al., 2023a) aim to extend the context size of Transformer-based models during fine-tuning, primarily by modifying positional encodings. For example, Zhang et al. (2024b) introduces a novel extension to RoPE (Su et al., 2024) which combines adjusting RoPEs base frequency and scaling the attention logits to help LLMs efficiently adapt to a larger context window. Chen et al. (2024) generalizes the positional encoding scaling approaches to model the continuous dynamics by ordinary differential equations over the length scaling factor. (Chen et al., 2023a) proposes to extend the context length by slightly modifying RoPE via Position Interpolation (PI) and fine-tuning on a small amount of data. Our approach, in contrast, focuses on the model's output space, which identifies the crucial role of output alignment in length generalization. Wang et al. (2024) proposes a novel approach designed to narrow the generalization gap by refining the interpolation of RoPE features for OOD positions and provides length extrapolation analysis on the feature gap.

## 3 A CASE STUDY ON SYNTHETIC TASKS: HOW OUTPUT ALIGNMENT AFFECTS LENGTH GENERALIZATION?

Length generalization refers to the ability to generalize from smaller training context sizes to larger ones. Previous research has explored various factors critical to successful length generalization, such as the task type (Zhou et al., 2024; Jelassi et al., 2023; Nogueira et al., 2021) and the positional encoding (Ontanon et al., 2022; Kazemnejad et al., 2023; Ruoss et al., 2023). However, in this work, we propose a new perspective, identifying output distribution alignment across different positions as another important factor.

**Mean Prediction v.s. Length Prediction.** We start from a case study on synthetic tasks: in the mean prediction task, the prediction target is the mean value of the sequence, while in the length prediction task, the target is the length of the sequence. We focus on binary input sequences, where each position in the sequence is filled with $0$ or $1$ with the same probability, and the decoder-only Transformer (Vaswani et al., 2017), a model widely used in both synthetic tasks (Zhou et al., 2024;

Jelassi et al., 2023) and LLMs (Touvron et al., 2023; Peng et al., 2024), which utilizes a causal mask in the self-attention module to enable auto-regressive generation. More model details can be found in Appendix A. We train the model on sequences with a maximum length of $l_{\text{train}} = 10$ and test it on sequences with a maximum length of $l_{\text{test}} = 50$. Figure 1a and Figure 1b display the test results for both tasks. We observe that regardless of the positional embedding used, the test loss of the length prediction task dramatically increases when the test sequence length exceeds 10, the maximum training length. Furthermore, the test loss continues to rise as the test sequence length grows, indicating that the model demonstrates very **low length generalization ability in the length prediction task.** In contrast, the model exhibits **strong generalization ability in the mean prediction task**, as the test loss on longer sequences remains nearly consistent with the loss on shorter sequences. We provide a theoretical analysis of this observation in the Appendix B.1.

From the results above, it is evident that while the mean and length of a sequence all convey global information, the model's length generalization ability varies across these tasks. A key distinction lies in the differences in **output distribution** for each task. In the mean prediction task, where the model generalizes well, the output remains within the fixed range of $[0, 1]$, regardless of sequence length. However, in the length prediction task, where generalization is poor, the support set of the output distribution shifts to a single-point set $\{l\}$ as the sequence length increases to $l$. This distinction between the two types of tasks motivates us to consider the importance of the alignment in the output distribution for better length generalization ability.

**Explicit output alignment helps length generalization.** We propose OutRep, a reparameterization technique to explicitly improve output distribution alignment in synthetic tasks, thereby enhancing the model's length generalization ability. In the length prediction task, the output distribution for sequences of certain lengths is known. Leveraging this prior knowledge, during training, we apply a reversible function $f : \mathbb{R} \to \mathbb{R}$ to map the support sets of output distributions for sequences of varying lengths into more aligned sets. Instead of using the original target $y(\mathbf{x})$, we train the model on the transformed target $f(y(\mathbf{x}))$. At test time, we apply the reverse function $f^{-1}$ to the output to recover the original prediction. This approach aligns the output distributions across different lengths, which is expected to improve length generalization. We consider the following reparameterization functions: $f(x) = \sqrt{x}$, $f(x) = \log(x)$ and $f(x) = 1/\sqrt{x}$. We show the experiment results in Figure 1c. It can be observed that all three reparameterization functions successfully relieve the poor length generalization ability in the length prediction task. Specifically, the reparameterization function $f(x) = 1/\sqrt{x}$ has a nearly perfect generalization ability when the length is no more than 35. The rising trend when the test sequence length becomes longer is still slow. These results verify our conjecture on the output alignment that **better output alignment leads to improved length generalization ability**. We add more theoretical results and discussions in Appendix B and Appendix C. In the next section, we will extend these findings to the more practical natural language tasks.

## 4 OUTPUT ALIGNMENT IN NATURAL LANGUAGE TASKS

In the previous section, we observed a positive correlation between length generalization ability and output distribution alignment in synthetic tasks. Motivated by this finding, in this section, we aim to extend this investigation to natural language tasks. First, we introduce a metric to quantify output distribution alignment in sequence modeling and demonstrate its strong correlation with performance on long-context benchmarks. Building on this insight, we propose incorporating this metric as a regularization loss during training to improve output alignment, which can lead to the performance gains detailed in Section 5.

### 4.1 LONG-SHORT MISALIGNMENT: QUANTIFYING OUTPUT ALIGNMENT OF LANGUAGE MODELS

In synthetic tasks, we measure the discrepancy between the support sets of output distributions to capture the differences in output across varying sequence lengths. However, in natural language tasks, the model output is a vector $g_\theta(\mathbf{x}) \in \mathbb{R}^{|\mathcal{V}|}$, where the dimension is the size of the vocabulary $|\mathcal{V}|$. This makes it challenging to directly apply the same analysis from synthetic tasks to natural language tasks. Despite this, similar output misalignment issues can still be observed in natural language tasks. Specifically, for a sequence $\mathbf{x}$ and its suffixes $\mathbf{x}_{[-l_1:]}$ and $\mathbf{x}_{[-l_2:]}$, where $l_1$ and $l_2$

are two lengths and $\mathbf{x}_{[-l_i:]}$ means the last $l_i$ tokens of $\mathbf{x}$ ($i = 1, 2$), the model's output is expected to remain consistent when $l_1$ and $l_2$ are similar, because the two suffixes share large overlap in tokens, resulting in similar contextual information. However, we find that models with poor length generalization tend to produce distant output distributions when conditioned on these sequences.

### 4.1.1 METRIC

To quantitatively explore the relationship between output distribution alignment and length generalization ability, we want to first design a metric to evaluate output alignment. We propose to utilize symmetrical cross-entropy (SCE) loss (Wang et al., 2019) to measure the divergence between output distributions conditioned on two distinct sequences. Consider two input sequences, $\mathbf{x}$ and $\mathbf{x}'$ with corresponding model predictions $\mathbf{y} = g_\theta(\mathbf{x})$ and $\mathbf{y}' = g_\theta(\mathbf{x}')$. The SCE loss between these predictions is defined as:

$$\mathcal{L}_{\mathrm{SCE}}(\mathbf{y}, \mathbf{y}') = -\left(\langle \mathbf{y}', \log(\mathbf{y}) \rangle + \langle \mathbf{y}, \log(\mathbf{y}') \rangle\right), \tag{1}$$

where $\langle \cdot, \cdot \rangle$ denotes the inner product and the $\log$ function is applied element wise. A lower SCE loss between the two predictions indicates better alignment. To assess overall output distribution alignment, we compute the expectation over sequence lengths $l_1$ and $l_2$ for a given input $\mathbf{x}$:

$$\mathcal{L}_{\mathrm{misalign}}(g_\theta) = \mathbb{E}_{\mathbf{x}, l_1, l_2} \mathcal{L}_{\mathrm{SCE}}(g_\theta(\mathbf{x}_{[-l_1:]}), g_\theta(\mathbf{x}_{[-l_2:]})). \tag{2}$$

We refer to this metric as the **long-short misalignment**, where a lower value signifies better output alignment across different sequence lengths. An illustration of this metric is shown in Figure 2a. In practice, we sample $l_1$ and $l_2$ from the interval $[l_{\mathrm{train}}/2, l_{\mathrm{train}}]$, where $l_{\mathrm{train}}$ represents the maximum context length used during training.

### 4.1.2 RESULTS

To evaluate the model's length generalization ability, we use the perplexity on long validation sets (16k length) and the LongBench-E score (Bai et al., 2023b). For perplexity evaluation, we select a subset from the RedPajama-Book corpus (Computer, 2023), following the protocol in (Chen et al., 2024). LongBench-E is a multitask benchmark that comprehensively evaluates large language models' ability to understand long contexts, with task lengths averaging between 5k and 32k tokens.

**Empirical Results.** Table 1 shows the long-short misalignment metric $\mathcal{L}_{\mathrm{misalign}}$ for the models along with their corresponding long-context evaluation results. Additionally, we include the training loss $\mathcal{L}_{\mathrm{train}}$ for each model on the RedPajama-Book corpus, which reflects the perplexity on sequences with the maximum training length $l_{\mathrm{train}}$. Interestingly, while the training loss (i.e., log of perplexity on sequences of length $l_{\mathrm{train}}$) shows a moderate correlation with long-context performance metrics—indicating that lower training loss can contribute to improved length generalization—the long-short misalignment metric $\mathcal{L}_{\mathrm{misalign}}$ demonstrates a much stronger correlation with long-context performance, as evidenced by its higher absolute correlation coefficient. These findings suggest that $\mathcal{L}_{\mathrm{misalign}}$ is a promising indicator of length generalization ability. However, it is important to note that we do not claim any causal relationship based solely on these observations. We will elaborate more on this relationship in Section 5.

Additionally, we also provide theoretical support for this observation, extending previous work on autoregressive modeling (Zhang et al., 2024a) with a theorem:

**Theorem 1** (*Generalization guarantees for the natural language task*). *Under some model assumptions, the generalization error $\mathcal{E}_{\mathrm{gen}}(g_\theta; l_{\mathrm{test}})$ with testing length $l_{\mathrm{test}}$ is upper bounded by the sum of training loss $\mathcal{L}_{\mathrm{train}}(g_\theta)$ and misalignment metric $\mathcal{L}_{\mathrm{misalign}}(g_\theta)$, i.e.,*

$$\mathcal{E}_{\mathrm{gen}}(g_\theta; l_{\mathrm{test}}) \leq C_1^{(l_{\mathrm{test}})} \cdot \mathcal{L}_{\mathrm{misalign}}(g_\theta) + C_2^{(l_{\mathrm{test}})} \cdot \mathcal{L}_{\mathrm{train}}(g_\theta) + C_0^{(l_{\mathrm{test}})}, \tag{3}$$

*where $C_i^{(l_{\mathrm{test}})}$ are constants related to $l_{\mathrm{test}}$. Specifically, the ratio $C_1^{(l_{\mathrm{test}})}/C_2^{(l_{\mathrm{test}})}$ becomes larger as $l_{\mathrm{test}}$ increase. This indicates that as the testing length increases, the alignment loss becomes increasingly significant.*

This theorem highlights the importance of minimizing both $\mathcal{L}_{\mathrm{misalign}}$ and $\mathcal{L}_{\mathrm{train}}$ to achieve lower generalization error. Moreover, as the testing length $l_{\mathrm{test}}$ increases, reducing $\mathcal{L}_{\mathrm{misalign}}$ plays a more critical role in improving generalization performance. The above empirical and theoretical results motivate us to explicitly optimize the long-short misalignment metric to enhance the model's ability to handle long-context sequences, which will be stated in the following sections.

Table 1: The proposed long-short misalignment metric $\mathcal{L}_{\text{misalign}}$ of models, with their log of perplexity (PPL) on 16k-long contexts and LongBench-E score. We also provide $\mathcal{L}_{\text{train}}$ as an additional metric in comparison. We find that $\mathcal{L}_{\text{misalign}}$ correlates better with the long-context benchmark performance.

| Model | $\mathcal{L}_{\text{train}}$ | $\mathcal{L}_{\text{misalign}}$ | log(PPL) | LongBench-E Score |
|---|---|---|---|---|
| GPT-J-6B (Wang, 2021) | 2.1 | 3.4 | 9.5 | 7.8 |
| GPT-NeoX-20B (Black et al., 2022) | 2.3 | 3.2 | 9.4 | 9.7 |
| Llama2-7B (Touvron et al., 2023) | 1.9 | 3.4 | 9.4 | 8.9 |
| RandomPos (Ruoss et al., 2023) | 2.2 | 3.3 | 8.2 | 9.2 |
| Yarn-Llama-2-7B-8k (Peng et al., 2024) | 2.0 | 2.6 | 3.8 | 21.2 |
| Qwen-7B-8k (Bai et al., 2023a) | 1.7 | 2.8 | 3.2 | 24.2 |
| CLEX-LLaMA-4K (Chen et al., 2024) | 1.7 | 2.4 | 1.8 | 32.7 |

| Metric | | Correlation Coefficient with Metric | |
|---|---|---|---|
| $\mathcal{L}_{\text{train}}$ | | 0.67 | -0.59 |
| $\mathcal{L}_{\text{misalign}}$ | | 0.88 | -0.95 |

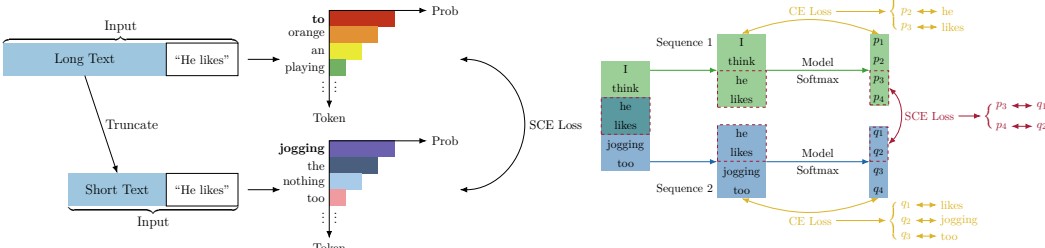

(a) Illustration of long-short misalignment metric $\mathcal{L}_{\text{misalign}}$.

(b) Illustration of efficiently calculating the total training loss $\mathcal{L}_{\text{train}}^*$.

Figure 2: (a) Given two input sequences, where one is a truncated version of the other, the long-short misalignment metric is computed by taking the expectation on Symmetrical Cross-Entropy (SCE) loss (Wang et al., 2019) between the model's predictions for these two sequences. (b) Illustration of efficiently calculating the total training loss. This implementation requires only two forward propagations for the two sequences, resulting in minimal additional time and resource costs

## 4.2 LONG-SHORT MISALIGNMENT METRIC AS REGULARIZATION LOSS

Since both empirical and theoretical results indicate a strong correlation between the proposed long-short misalignment metric and long-context benchmark performance, we incorporate this metric as a regularization loss into the training loss, resulting in the new training loss defined as:

$$\mathcal{L}_{\text{train}}^*(g_\theta) = \mathcal{L}_{\text{train}}(g_\theta) + \alpha \cdot \mathcal{L}_{\text{misalign}}(g_\theta), \tag{4}$$

where $\mathcal{L}_{\text{train}}$ is the original cross-entropy training loss and $\alpha$ is the regularization coefficient. Calculating these two losses separately during training can be time-consuming, as the computation of $\mathcal{L}_{\text{misalign}}$ requires forward propagation through two distinct sequences. To address this, we propose an efficient implementation for $\mathcal{L}_{\text{train}}^*$. We first sample an integer $l_{\text{extra}}$ from $[1, l_{\text{train}}/2]$ and then sample a sequence of length $l_{\text{train}} + l_{\text{extra}}$. The first $l_{\text{train}}$ tokens form the first input sequence, while the last $l_{\text{train}}$ tokens form the second input sequence. Both sequences can be used to compute $\mathcal{L}_{\text{train}}$. The overlap between the two sequences starts at token $l_{\text{extra}} + 1$ and continues to token $l_{\text{train}}$, resulting in an overlap of $l_{\text{train}} - l_{\text{extra}}$ tokens. We can calculate the long-short misalignment loss in the overlapping positions. This implementation requires only two forward propagations for the two sequences, resulting in minimal additional time and resource costs compared to calculating the original train loss $\mathcal{L}_{\text{train}}$. A detailed Pytorch-like algorithm is provided in Appendix D and an overall illustration can be found in Figure 2b. We will conduct experiments using the proposed regularization loss in the next section to show its effectiveness.

Table 2: Performance of the fine-tuned models using only cross-entropy loss (baseline) and an additional long-short misalignment loss on long-context modeling benchmark, LongBench-E score (Bai et al., 2023b) and perplexity on the 8k-length validation set. The fine-tuning sequence length is 4k, exactly the same as the training sequence length. We adopt two datasets: RedPajama-Book (Computer, 2023) and PG19 (Rae et al., 2019). The models finetuned with our proposed loss outperform the baseline across different model adaption strategies.

| Benchmark | LongBench-E ($\uparrow$) | | | Perplexity ($\downarrow$) | | |
|---|---|---|---|---|---|---|
| Training steps | 50 | 100 | 200 | 50 | 100 | 200 |
| *RedPajama-Book* | | | | | | |
| $\mathcal{L}_{\text{train}}$ (Baseline) | 22.7 | 23.8 | 24.7 | 7.21 | 6.56 | 6.12 |
| $\mathcal{L}_{\text{train}} + 0.1\mathcal{L}_{\text{misalign}}$ (Ours) | **23.1** | **25.2** | **26.6** | **6.89** | **6.24** | **5.88** |
| $\mathcal{L}_{\text{train}} + 0.5\mathcal{L}_{\text{misalign}}$ (Ours) | 21.9 | 23.7 | 24.7 | 7.44 | 7.01 | 6.54 |
| *PG19* | | | | | | |
| $\mathcal{L}_{\text{train}}$ (Baseline) | 20.2 | 21.4 | 22.5 | **8.92** | **7.89** | 7.45 |
| $\mathcal{L}_{\text{train}} + 0.1\mathcal{L}_{\text{misalign}}$ (Ours) | **20.7** | 22.1 | **25.3** | 8.95 | 7.92 | **7.35** |
| $\mathcal{L}_{\text{train}} + 0.5\mathcal{L}_{\text{misalign}}$ (Ours) | 20.1 | **22.2** | 23.6 | 9.42 | 8.59 | 8.21 |

## 5 Experiments on Natural Language Tasks

In this section, we conduct extensive experiments to verify the effectiveness of our proposed length alignment loss. We first examine the proposed output distribution alignment loss on length generalization tasks, where the model is trained on short sequences and tested on longer ones. Additionally, we explore its application in another common scenario: long-context learning, where both training and testing involve long sequences.

### 5.1 Experiments with Training on Short Sequences

In this section, we consider the classical length generalization setting, where the model is trained on short sequences (4k-long) and tested on longer sequences (at least 5k-long). Due to the high computational cost of pre-training large language models from scratch, most current methods fine-tune open-sourced pre-trained models (Chen et al., 2024; Yang, 2023; Peng et al., 2024). In our experiments, we use Llama2-7b (Touvron et al., 2023) as the base model and apply the CLEX (Chen et al., 2024) adjustment method. We use two datasets: the RedPajama-Book corpus (Computer, 2023) and PG19 (Rae et al., 2019). The experiments are conducted with a context length of 4,096, a batch size of 64, and a maximum of 200 training steps. For the regularization coefficient $\alpha$, we test values of 0.1 and 0.5.

We evaluate performance using the LongBench-E score (Bai et al., 2023b) and perplexity on validation sets made up of sequences of length 8,192 from the corpus of the respective training dataset. LongBench-E is a multitask benchmark that comprehensively evaluates large language models' ability to understand long contexts, with task lengths averaging between 5k and 32k tokens, which has been adopted by many previous works (Chen et al., 2024; Jin et al., 2024) as an effective evaluation metric for long-context modeling. The results, shown in Table 2, indicate that the model fine-tuned with our proposed loss consistently outperforms the baseline model on the LongBench-E benchmark. The fine-tuned model shows lower perplexity on RedPajama-Book and similar perplexity on PG19. These results support the effectiveness of our proposed loss and the intuition that lower misalignment metric $\mathcal{L}_{\text{misalign}}$ leads to better length generalization ability. For the regularization coefficient $\alpha$, we find that larger values do not always improve performance, as they may interfere with the models next-word prediction.

### 5.2 Experiments with Training on Longer Sequences

In this section, we consider the scenario that the model is finetuned on a longer sequence than the training length. We use different model adjustment strategies during the fine-tuning stage, to demonstrate that the proposed length alignment loss can be applied to various long-context fine-tuning methods. Our experiments use Llama2-7b as the base model. For model adjustments, we consider

Table 3: Performance of the finetuned models using only cross-entropy loss (baseline) and an additional long-short misalignment loss on long-context modeling benchmark, LongBench-E score (Bai et al., 2023b) and perplexity on the 8k-length validation set. The fine-tuning sequence length is 8k. We adopt two kinds of model adjustments: LongQLora (Yang, 2023) and EABF (Zhang et al., 2024b). The models finetuned with our proposed loss outperform the baseline across different model adaption strategies.

| Benchmark | LongBench-E (↑) | | | Perplexity (↓) | | |
|---|---|---|---|---|---|---|
| Training steps | 50 | 100 | 200 | 50 | 100 | 200 |
| *LongQLora* | | | | | | |
| $\mathcal{L}_{\text{train}}$ (Baseline) | **21.9** | 22.1 | 23.4 | 6.82 | 6.41 | 5.82 |
| $\mathcal{L}_{\text{train}} + 0.1\mathcal{L}_{\text{misalign}}$ (Ours) | 21.8 | 23.3 | **25.8** | **6.72** | **6.39** | **5.77** |
| $\mathcal{L}_{\text{train}} + 0.5\mathcal{L}_{\text{misalign}}$ (Ours) | 21.4 | **23.9** | 25.1 | 7.07 | 6.62 | 5.92 |
| *EABF* | | | | | | |
| $\mathcal{L}_{\text{train}}$ (Baseline) | 22.1 | 22.9 | 23.6 | **6.89** | 6.52 | 6.01 |
| $\mathcal{L}_{\text{train}} + 0.1\mathcal{L}_{\text{misalign}}$ (Ours) | **23.2** | **24.0** | 24.8 | 6.92 | **6.43** | **5.91** |
| $\mathcal{L}_{\text{train}} + 0.5\mathcal{L}_{\text{misalign}}$ (Ours) | 22.5 | 23.2 | 23.9 | 7.14 | 6.78 | 6.34 |

two approaches: LongQLora (Yang, 2023) and EABF (Zhang et al., 2024b). LongQLora leverages multiple techniques, including Position Interpolation (Chen et al., 2023a), QLoRA (Dettmers et al., 2024), and Shift Short Attention from LongLoRA (Chen et al., 2023b). Meanwhile, EABF introduces a dynamic rescaling mechanism to the attention layers and applies a higher base frequency for RoPE. The experiments are conducted on the RedPajama-Book corpus (Computer, 2023), with a context length of 8,192, a batch size of 64, and a maximum of 200 training steps. For the regularization coefficient $\alpha$, we test values of 0.1 and 0.5.

We evaluate performance using the LongBench-E score and perplexity on validation sets composed of sequences of length 8,192 from the RedPajama-Book corpus. The results are shown in Table 3. Notably, our methods outperform the baseline across both model adjustment strategies. Specifically, models trained with our proposed regularization loss achieve up to a 2.4% improvement in the LongBench-E score. Similar to the previous experiments, we observe that excessively large regularization coefficient values may not consistently benefit long-context modeling, which is reflected in the slightly lower LongBench-E scores and higher perplexity, indicating that overly strong regularization may disrupt the model's training process.

## 5.3 ABLATION STUDIES

Since we incorporate several hyperparameters such as the regularization coefficient $\alpha$ and the sampling range $|l_1 - l_2|$ in the misalignment metric, we conduct extensive experiments to explore how these hyperparameters affect the model performance.

**Regularization coefficient $\alpha$.** In addition to the settings already provided in the previous experiments ($\alpha = 0$ as the baseline, $\alpha = 0.1$, and $\alpha = 0.5$), we evaluated $\alpha = 0.3$ and $\alpha = 1.0$ under the same experimental conditions as Table 2, using CLEX as the model adjustment. We still adopt RedPajama-Book as the training dataset. The results are shown in Table 4, which reveal the following trend: (1) Performance peaks for $\alpha$ values in the range $[0.1, 0.3]$ in both evaluation metrics. (2) Larger values of $\alpha$ (e.g., $\alpha = 0.5$ or $\alpha = 1.0$) lead to a significant decline in performance, confirming the risks of over-regularization. These findings highlight the importance of selecting a moderate value for $\alpha$. We suggest using a coefficient $\alpha$ between 0.1 and 0.3 as default to mitigate the risk of over-regularization.

**Sampling range.** In equation 2, we sample $l_1$ and $l_2$ from $[l_{\text{train}}/2, l_{\text{train}}]$ by default to avoid input sequence with significantly different lengths. This is equivalent to sampling $l_{\text{extra}}$ from $[1, l_{\text{train}}/2]$. Here we conduct an ablation study examining how different sampling strategies of $l_{\text{extra}}$ affect performance. We consider four sampling configurations: (1) The current strategy, sampling from $[1, l_{\text{train}}/2]$; (2) Sampling from a narrower range $[1, l_{\text{train}}/4]$; (3) Sampling from a narrower range $[l_{\text{train}}/4, l_{\text{train}}/2]$; (4) Sampling from a broader range $[1, l_{\text{train}}]$ and remove the limit that $l_1$ and $l_2$ should be in $[l_{\text{train}}/2, l_{\text{train}}]$. We conduct experiments using the same setting as Table 3, using

Table 4: Ablation study on the regularization coefficient $\alpha$. The setting is the same as Table 2. We adopt RedPajama-Book (Computer, 2023) as the training dataset. We find it important to select a moderate value for $\alpha$.

| Benchmark | LongBench-E ($\uparrow$) | | | Perplexity ($\downarrow$) | | |
|---|---|---|---|---|---|---|
| Training steps | 50 | 100 | 200 | 50 | 100 | 200 |
| *RedPajama-Book* | | | | | | |
| $\mathcal{L}_{\text{train}}$ (Baseline) | 22.7 | 23.8 | 24.7 | 7.21 | 6.56 | 6.12 |
| $\mathcal{L}_{\text{train}} + 0.1\mathcal{L}_{\text{misalign}}$ | 23.1 | 25.2 | 26.6 | **6.89** | **6.24** | **5.88** |
| $\mathcal{L}_{\text{train}} + 0.3\mathcal{L}_{\text{misalign}}$ | **23.4** | **25.8** | **27.1** | 6.95 | 6.35 | 5.98 |
| $\mathcal{L}_{\text{train}} + 0.5\mathcal{L}_{\text{misalign}}$ | 21.9 | 23.7 | 24.7 | 7.44 | 7.01 | 6.54 |
| $\mathcal{L}_{\text{train}} + 1.0\mathcal{L}_{\text{misalign}}$ | 18.2 | 19.4 | 19.9 | 16.21 | 14.12 | 12.92 |

Table 5: Ablation study on the regularization coefficient $\alpha$. The setting is the same as Table 3. We adopt RedPajama-Book (Computer, 2023) as the training datasets and LongQLora (Yang, 2023) as the model adjustment method. We find it important to carefully balance the sampling range to optimize the model's generalization to longer contexts.

| Benchmark | LongBench-E ($\uparrow$) | | | Perplexity ($\downarrow$) | | |
|---|---|---|---|---|---|---|
| Training steps | 50 | 100 | 200 | 50 | 100 | 200 |
| *RedPajama-Book* | | | | | | |
| (1) Sampling $l_{\text{extra}}$ from $[1, l_{\text{train}}/2]$ (Current) | **21.8** | **23.3** | **25.8** | **6.72** | 6.39 | **5.77** |
| (2) Sampling $l_{\text{extra}}$ from $[1, l_{\text{train}}/4]$ | 21.5 | 23.2 | 25.7 | 6.77 | **6.29** | 5.81 |
| (3) Sampling $l_{\text{extra}}$ from $[l_{\text{train}}/4, l_{\text{train}}/2]$ | 21.4 | 22.7 | 24.5 | 6.82 | 6.47 | 5.94 |
| (4) Sampling $l_{\text{extra}}$ from $[1, l_{\text{train}}]$ | 18.2 | 18.9 | 19.1 | 15.65 | 13.59 | 12.52 |

LongQLora for model adjustments and a regularization coefficient of 0.1. The results are shown in Table 5: (1) Setting 2 achieved performance comparable to the current strategy, while Setting 3 showed slightly inferior performance compared to the current strategy. This suggests that **aligning outputs between sequences with moderate length discrepancies effectively supports long-context modeling.** (2) Setting 4 yielded significantly worse performance than the current strategy, indicating that **encouraging alignment between sequences with large length differences adversely affects the model's long-context capabilities.** These results underscore the importance of carefully balancing the sampling range to optimize the model's generalization to longer contexts.

## 5.4 EXPERIMENTS ON BABILONG

We conduct extensive experiments on BABILong (Kuratov et al., 2024), a challenging reasoning-in-a-haystack task specifically designed for evaluating long-context capabilities. BABILong comprises question-answering tasks where the supporting facts for each question are situated at specific positions within the context. Using the model setup described in Section 5.1, which incorporates CLEX as the adjustment method and RedPajama-Book as the training dataset, all models are fine-tuned for 200 steps. Evaluation is performed on input sequences of lengths 4K, 8K, and 16K, with the overall results summarized in Table 6. The results indicate that our proposed method consistently outperforms the baseline across all evaluated lengths. Specifically, our method achieves a performance gain of 2.0% at length 8K and 2.2% at length 16K. These results demonstrate the effectiveness of our regularization loss in enhancing length generalization.

Additionally, we analyze the impact of the supporting fact's position within the input context using the QA1 task from BABILong, where each question is associated with a single supporting fact. Results from this analysis are presented in Table 7, offering two key insights: (1) **Performance with early-context facts:** When the supporting fact is located at the beginning of the input context (fact depth = 0), our method achieves performance comparable to the baseline. This suggests that, despite the form of the regularization potentially encouraging the model to neglect earlier contexts, it does not lead to this behavior in practice. (2) **Performance with middle-context facts:** When the supporting fact is positioned in the middle of the context (fact depth = 50 or 75), our method shows

Table 6: The overall evaluation results on BABILong (Kuratov et al., 2024) with sequence lengths of 4K, 8K, 16K.

| Training Loss | Evaluation Length | | |
|---|---|---|---|
| | 4K | 8K | 16K |
| $\mathcal{L}_{\text{train}}$ (Baseline) | 48.2 | 42.4 | 37.9 |
| $\mathcal{L}_{\text{train}} + 0.1\mathcal{L}_{\text{misalign}}$ | **49.1** | **44.4** | **40.1** |

Table 7: The evaluation results on BABILong with different locations of the facts in the QA1 task. Input length is 16K.

| Training Loss | Fact Depth (%) | | | |
|---|---|---|---|---|
| | 0 | 25 | 50 | 75 |
| $\mathcal{L}_{\text{train}}$ (Baseline) | **75** | 64 | 30 | 69 |
| $\mathcal{L}_{\text{train}} + 0.1\mathcal{L}_{\text{misalign}}$ | 73 | 64 | **38** | **74** |

Table 8: Comparison between synthetic tasks and natural language tasks.

| | Synthetic Tasks | Language Tasks |
|---|---|---|
| Output Space | $\mathbb{R}$ | $L_1$ unit ball in $\mathbb{R}^{|\mathcal{V}|}$ |
| Specific Task | Length/Sum prediction | Next token prediction |
| Output Distribution Misalignment | Exist | Exist |
| Priori on Output Distribution | Explicit and predifined | Implicit and task-dependent |
| Alignment Technique | Explicit reparameterization | Regularization loss across lengths |
| Does the technique decrease output distribution misalignment? | Yes (explicitly) | Yes (implicitly through optimization) |
| Does the technique improve length generalization? | Yes | Yes |

considerable improvement over the baseline. This indicates that our approach effectively mitigates the "loss-in-the-middle" phenomenon (Liu et al., 2024), a common challenge in large language models. Together, these results strongly support the effectiveness of our proposed regularization loss in enhancing length generalization ability, particularly for tasks requiring attention across diverse positions.

# 6 DISCUSSION

Since our work is initially motivated by phenomena observed in synthetic tasks, we provide additional clarification on the relationship between synthetic tasks and natural language tasks by summarizing their key differences and similarities in Table 8. While these two types of tasks differ significantly in their specific forms and output space, they share a common challenge: **output distribution misalignment across different input lengths**. Our analysis highlights that employing an alignment technique–whether explicit reparameterization in synthetic tasks or regularization in natural language tasks–can effectively mitigate this misalignment. This mitigation directly enhances the model's length generalization ability, demonstrating the broader applicability of our approach.

# 7 CONCLUSION

In this work, we focused on length generalization tasks and introduced a new perspective by examining the output space. We first identified the critical role of output alignment in length generalization, demonstrating both empirically and theoretically that **output distribution misalignment across different input sequence lengths leads to poor length generalization**. Building on this insight, we proposed a reparameterization technique, OutRep, to align the output space, and confirmed its effectiveness through empirical and theoretical validation. We extended this approach to natural language tasks, introducing a metric called **Long-Short Misalignment** to quantify output alignment, which showed a strong correlation with length generalization performance. Based on these findings, we proposed a regularization loss during training to improve output alignment. Extensive experiments further validated the effectiveness of this approach. Overall, our work offers a novel perspective for understanding and enhancing length generalization in large language models.

## 8 ETHICS STATEMENT

We believe there are no direct ethical problems in this work since it primarily focuses on the theoretical analysis and improvement of large language models (LLMs). However, LLMs can generate inaccurate or harmful content, and this research does not offer a direct solution to these issues. Users are encouraged to ensure that the LLMs obey the ethical standards when implementing the proposed methods.

## 9 REPRODUCIBILITY STATEMENT

We have provided sufficient materials to ensure reproducibility. The details of experiments including training settings, models and datasets are provided in Section 5, Appendix A and Appendix D. The formal statement and complete proofs of theorems are provided in Appendix B.

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

## A  MODEL DETAILS FOR SYNTHETIC TASKS

We focus on the decoder-only Transformer (Vaswani et al., 2017), a model widely used in both synthetic tasks (Zhou et al., 2024; Jelassi et al., 2023) and LLMs (Touvron et al., 2023; Peng et al., 2024) which utilizes a causal mask in the self-attention module to enable auto-regressive generation. We consider several positional encodings: learnable positional encoding (Radford et al., 2019), Alibi (Press et al., 2021), rotary positional encoding (Su et al., 2024) and no positional encoding (NoPE). Since recent works found that by removing the positional encoding, Transformers can trained to be well generalized on length (Deletang et al., 2022; Kazemnejad et al., 2023), we adopt this setting (i.e. NoPE) by default. To provide a clear signal for the model to accomplish the tasks, we add both the begin-of-sentence (BOS) token and the end-of-sentence (EOS) token in the sequence. We apply a feed-forward neural network on the hidden state of the last token to generate an output of a real number. We train all of our models on the train distribution from scratch to convergence if possible. For all tasks, the length of training examples is sampled uniformly from length 1 up to the max training length $l_{\text{train}}$. We select hyper-parameters such as the learning rate for each task based on what is required to fit the training set. At test time, the length of the examples will traverse from 1 to the max testing length $l_{\text{test}}$.

## B  THEOREMS AND PROOFS

### B.1  THEORETICAL ANALYSIS FOR SYNTHETIC TASKS

Following (Zhang et al., 2024a), our analysis is based on the linear attention model. The general form of linear attention is given by:

$$\text{Attn}(\mathbf{x}) = QK^\top V = \mathbf{x}W^Q(\mathbf{x}W^K)^\top \mathbf{x}W^V, \tag{5}$$

where $W^Q, W^K, W^V$ are projections, and $n$ is the length of input $\mathbf{x}$. In all tasks, we will use a linear attention model with a causal mask. Specifically, we normalize the output according to its position, which allows linear attention to perform similarly to dot-product attention. For example, the $k$-th output will be normalized as follows:

$$\frac{1}{k}Q_k K^\top V = \frac{1}{k}\sum_{i=1}^{k} Q_k K_i^\top V_i. \tag{6}$$

For the synthetic tasks, we add bias terms to $Q, K, V$ to mitigate the impact of 0 in the input. The model based on linear attention is defined as follows:

$$g_\theta(\mathbf{x}_{[:k]}) = \frac{1}{k}\sum_{i=1}^{k} Q_k K_i^\top V_i, \tag{7}$$

and it is trained based on the following target function:

$$\mathcal{L}(g_\theta; l_{\text{train}}) = \mathbb{E}_{\mathbf{x}\in\{0,1\}^{l_{\text{train}}}} \frac{1}{l_{\text{train}}}\sum_{i=1}^{l_{\text{train}}} \left\| g_\theta(\mathbf{x}_{[:i]}) - y(\mathbf{x}_{[:i]}) \right\|_2^2, \tag{8}$$

where $\mathbf{x}_{[:i]}$ indicates the first $i$-tokens of input $\mathbf{x}$. The optimal model trained on sequence with maximum length of $l_{\text{train}}$ is denoted as $g_\theta^{l_{\text{train}}}$. We have the following result:

**Theorem 2.** *In the length prediction task and the sum prediction, the length generalization loss has a quadratic relationship with the predicted length, i.e.,*

$$\mathcal{E}_{\text{length}}(g_\theta^{l_{\text{train}}}; l_{\text{test}}) = \mathbb{E}_{\mathbf{x}_{\text{test}}\in\{0,1\}^{l_{\text{test}}}} \left\| g_\theta^{l_{\text{train}}}(\mathbf{x}_{\text{test}}) - y(\mathbf{x}_{\text{test}}) \right\|_2^2 = \mathcal{O}\left((l_{\text{test}} - l_{\text{train}})^2\right), \tag{9}$$

$$\mathcal{E}_{\text{sum}}(g_\theta^{l_{\text{train}}}; l_{\text{test}}) = \mathcal{O}\left((l_{\text{test}} - l_{\text{train}})^2\right). \tag{10}$$

*However, in the mean prediction task, the length generalization loss has a fixed upper bound:*

$$\mathcal{E}_{\text{mean}}(g_\theta^{l_{\text{train}}}; l_{\text{test}}) = \mathcal{O}(1). \tag{11}$$

*Proof.* Since the input consists of 0 and 1, the $Q_k, K_k, V_k$ will each have only two possible values. Thus, we may assume that when the input token is 1, the $Q_k, K_k, V_k$ are $q', k', v'$ respectively, and when the input token is 0, the $Q_k, K_k, V_k$ are $q'', k'', v''$ respectively. Furthermore, we note that in equation 7, $K_i$ and $V_i$ always share the same subscript. Therefore, we can treat them as a single token and replace them with $\kappa$, where we define $\kappa' = k'v'$ and $\kappa'' = k''v''$. We can decompose equation 8 into $l_{\text{train}}$ separate part:

$$\mathcal{L}(g_\theta; l_{\text{train}}) = \frac{1}{l_{\text{train}}} \sum_{l=1}^{l_{\text{train}}} \ell(g_\theta; l), \quad \ell(g_\theta; l) = \mathbb{E}_{\mathbf{x}} \left\| g_\theta(\mathbf{x}_{[:l]}) - y(\mathbf{x}_{[:l]}) \right\|_2^2. \tag{12}$$

Next, we consider each task individually.

(a) First, we study the length prediction task. In this case, $y(\mathbf{x}_{[:l]}) = l$. Expanding $\ell(g_\theta; l)$ we have:

$$\ell(g_\theta; l) = \mathbb{E}_{\mathbf{x}} \left\| g_\theta(\mathbf{x}_{[:l]}) - y(\mathbf{x}_{[:l]}) \right\|_2^2$$
$$= \frac{1}{2^l} \sum_{i=0}^{l} C_{l-1}^{i-1} \left( \frac{1}{l}(iq'\kappa' + (l-i)q'\kappa'') - l \right)^2 + C_{l-1}^{i} \left( \frac{1}{l}(iq''\kappa' + (l-i)q''\kappa'') - l \right)^2.$$

It's easy to find that $\ell(g_\theta, l)$ achieves its minimum 0 if and only if $q', q'', \kappa', \kappa''$ satisfy the following conditions:

$$\begin{cases} q' = q'' \neq 0, \kappa' = \kappa'' \neq 0, \\ q'\kappa' = q''\kappa' = q'\kappa'' = q''\kappa'' = l. \end{cases} \tag{13}$$

These are also properties that the solution of $\partial\ell(g_\theta; l) = 0$ holds. Note that the first property holds for any $\ell(g_\theta; l)$ and is independent of $l$. Since $\mathcal{L}(g_\theta; l_{\text{train}})$ is composed of $\ell(g_\theta; l)$, and each solution of $\ell(g_\theta; l) = 0$ satisfying $q' = q'' \neq 0, \kappa' = \kappa'' \neq 0$, the global solution for $\partial\mathcal{L}(g_\theta; l_{\text{train}}) = 0$ should also have this property. Therefore, we may assume that $q'\kappa' = q''\kappa' = q'\kappa'' = q''\kappa'' = \Gamma$, and our goal is to find the $\Gamma$ that satisfying $\partial\mathcal{L}(g_\theta; l_{\text{train}})/\partial\Gamma = 0$, which means that this $\Gamma$ minimizes $\mathcal{L}(g_\theta; l_{\text{train}})$. In this case, it is easy to find that $\ell(g_\theta; l) = (\Gamma - l)^2$, which means that $\partial\ell(g_\theta; l)/\partial\Gamma = 2(\Gamma - l)$. Therefore, we have:

$$\frac{\partial\mathcal{L}(g_\theta; l_{\text{train}})}{\partial\Gamma} = \frac{2}{l_{\text{train}}} \sum_{l=1}^{l_{\text{train}}} (\Gamma - l), \tag{14}$$

solving $\partial\mathcal{L}(g_\theta; l_{\text{train}})/\partial\Gamma = 0$ and we have $\Gamma = (l_{\text{train}} + 1)/2$, so the $g_\theta^{l_{\text{train}}}$ satisfying $q'\kappa' = q''\kappa' = q'\kappa'' = q''\kappa'' = (l_{\text{train}} + 1)/2$. Therefore, we have:

$$\mathcal{E}_{\text{length}}(g_\theta^{l_{\text{train}}}; l_{\text{test}}) = \ell(g_\theta^{l_{\text{train}}}; l_{\text{test}}) = \left( l_{\text{test}} - \frac{l_{\text{train}} + 1}{2} \right)^2 = \mathcal{O}\left( (l_{\text{test}} - l_{\text{train}})^2 \right). \tag{15}$$

(b) We now study the sum prediction task. In this case, $y(\mathbf{x}_{[:l]}) = \sum_{i=1}^{l} \mathbf{x}_i$. Expanding $\ell(g_\theta; l)$ we have:

$$\ell(g_\theta; l) = \mathbb{E}_{\mathbf{x}} \left\| g_\theta(\mathbf{x}_{[:l]}) - y(\mathbf{x}_{[:l]}) \right\|_2^2$$
$$= \frac{1}{2^l} \sum_{i=0}^{l} C_{l-1}^{i-1} \left( \frac{1}{l}(iq'\kappa' + (l-i)q'\kappa'') - i \right)^2 + C_{l-1}^{i} \left( \frac{1}{l}(iq''\kappa' + (l-i)q''\kappa'') - i \right)^2.$$

It's easy to find that $\ell(g_\theta, l)$ achieves its minimum 0 if and only if $q', q'', \kappa', \kappa''$ satisfy the following conditions:

$$\begin{cases} q' = q'' \neq 0, \kappa' \neq 0, \kappa'' = 0, \\ q'\kappa' = q''\kappa' = l. \end{cases} \tag{16}$$

This is similar to the previous conditions equation 13. Thus, we can perform a similar analysis as above, and we similarly assume that $q'\kappa' = q''\kappa' = \Gamma$ with $\kappa'' = 0$. In this case, we have:

$$\ell(g_\theta; l) = \frac{1}{2^l l^2} \sum_{i=0}^{l} i^2 C_l^i (\Gamma - l)^2 = \frac{l+1}{2l}(\Gamma - l)^2. \tag{17}$$

Thus we have:

$$\frac{\partial \ell(g_\theta; l)}{\partial \Gamma} = \frac{l+1}{l}(\Gamma - l), \tag{18}$$

which leads to:

$$\frac{\partial \mathcal{L}(g_\theta; l_{\text{train}})}{\partial \Gamma} = \frac{1}{l_{\text{train}}} \sum_{l=1}^{l_{\text{train}}} \frac{\partial \ell(g_\theta; l)}{\partial \Gamma} = \frac{1}{l_{\text{train}}} = (l_{\text{train}} + H_{l_{\text{train}}})\Gamma - \frac{l_{\text{train}}(l_{\text{train}} + 3)}{2}, \tag{19}$$

where $H_n$ is the $n$-th harmonic number, i.e., $H_n = \sum_{i=1}^{n} 1/i$. Solving $\partial \mathcal{L}(g_\theta; l_{\text{train}})/\partial \Gamma = 0$ and we have $\Gamma = l_{\text{train}}(l_{\text{train}} + 3)/2(l_{\text{train}} + H_{l_{\text{train}}})$, so the $g_\theta^{l_{\text{train}}}$ satisfying $q'\kappa' = q''\kappa' = l_{\text{train}}(l_{\text{train}} + 3)/2(l_{\text{train}} + H_{l_{\text{train}}})$. Therefore, we have:

$$\begin{aligned}
\mathcal{E}_{\text{sum}}(g_\theta^{l_{\text{train}}}; l_{\text{test}}) = \ell(g_\theta^{l_{\text{train}}}; l_{\text{test}}) &= \frac{l_{\text{test}} + 1}{2l_{\text{test}}} \left( \frac{l_{\text{train}}(l_{\text{train}} + 3)}{2(l_{\text{train}} + H_{l_{\text{train}}})} - l_{\text{test}} \right)^2 \\
&\approx \frac{1}{2} \left( \frac{l_{\text{train}} + 3}{2 + \text{const}} - l_{\text{test}} \right)^2 \\
&= \mathcal{O}\left( (l_{\text{test}} - l_{\text{train}})^2 \right).
\end{aligned} \tag{20}$$

(c) Finally we study the mean prediction task. In this case, $y(\mathbf{x}_{[:l]}) = \sum_{i=1}^{l} \mathbf{x}_i/l$. Expanding $\ell(g_\theta; l)$ we have:

$$\begin{aligned}
\ell(g_\theta; l) &= \mathbb{E}_{\mathbf{x}} \left\| g_\theta(\mathbf{x}_{[:l]}) - y(\mathbf{x}_{[:l]}) \right\|_2^2 \\
&= \frac{1}{2^l} \sum_{i=0}^{l} C_{l-1}^{i-1} \left( \frac{1}{l}(iq'\kappa' + (l-i)q'\kappa'') - \frac{i}{l} \right)^2 + C_{l-1}^i \left( \frac{1}{l}(iq''\kappa' + (l-i)q''\kappa'') - \frac{i}{l} \right)^2.
\end{aligned}$$

It's easy to find that $\ell(g_\theta, l)$ achieves its minimum $0$ if and only if $q', q'', \kappa', \kappa''$ satisfy the following conditions:

$$\begin{cases} q' = q'' \neq 0, \kappa' \neq 0, \kappa'' = 0, \\ q'\kappa' = q''\kappa' = 1. \end{cases} \tag{21}$$

This is similar to the previous conditions equation 16. In fact, the above properties are irrelevant to $l$, so for all $\ell(g_\theta, l)$ these properties hold. Therefore, in this case, the optimal solution of $\mathcal{L}(g_\theta; l_{\text{train}})$ will satisfy equation 21, which leads to $\mathcal{L}(g_\theta; l_{\text{train}}) = 0$. Under this situation, the length generalization error is:

$$\mathcal{E}_{\text{sum}}(g_\theta^{l_{\text{train}}}; l_{\text{test}}) = \ell(g_\theta^{l_{\text{train}}}; l_{\text{test}}) = 0. \tag{22}$$

Consider the case where it is not optimal, i.e., when $q'\kappa' = q''\kappa'' = 1 + \varepsilon$, where $\varepsilon \neq 0$ is small, we can similarly obtain:

$$\mathcal{E}_{\text{sum}}(g_\theta^{l_{\text{train}}}; l_{\text{test}}) = \ell(g_\theta^{l_{\text{train}}}; l_{\text{test}}) = \frac{(l_{\text{test}} + 1)\varepsilon^2}{2l_{\text{test}}} \leq \varepsilon^2. \tag{23}$$

In conclusion, we have:

$$\mathcal{E}_{\text{sum}}(g_\theta^{l_{\text{train}}}; l_{\text{test}}) = \mathcal{O}(1), \tag{24}$$

which completes the proof.

$\square$

## B.2 THEORETICAL ANALYSIS FOR NATURAL LANGUAGE TASKS

We now focus on natural language tasks. We make some changes to the model following (Zhang et al., 2024a). In natural language tasks, the model requires an additional projection $W$ to make the output a probability distribution. That is, the model is modified as follows:

$$g_\theta(\mathbf{x}_{[:k]}) = \frac{1}{k} \sum_{i=1}^{k} Q_k K_i^\top V_i W, \tag{25}$$

In this case, each token $\mathbf{x}_i$, output $g_\theta(\mathbf{x})$ and the objective function $y(\mathbf{x})$ are normalized. Without loss of generality, we may assume that the changes in the objective function after truncating the inputs are negligible, i.e.

$$\mathbf{Pr}(y(\mathbf{x}_{[:l_1]}) \neq y(\mathbf{x}_{[:l_2]})) = 0 \quad (\forall l_1, l_2). \tag{26}$$

For simplicity, we use the $L_2$-norm instead of SCE to measure misalignment, i.e.:

$$\mathcal{L}_{\text{misalign}}(g_\theta) = \mathbb{E}_{\mathbf{x}, l_1, l_2} \|g_\theta(\mathbf{x}_{[-l_1:]}) - g_\theta(\mathbf{x}_{[-l_2:]})\|_2^2, \tag{27}$$

and we use the $L_2$-norm instead of the CE loss as the training loss function since these two functions differ only by a constant when the output and target are regularized. Under these conditions, we have the following result:

**Theorem 1** (*Generalization guarantees for the natural language task*). *Suppose that $g_\theta^{l_{\text{train}}}$ is the model trained on sequences with maximum training length $l_{\text{train}}$. When the testing length is $l_{\text{test}} > l_{\text{train}}$, the generalization loss $\mathcal{E}_{\text{gen}}(g_\theta^{l_{\text{train}}}; l_{\text{test}})$ has the following upper bound:*

$$\mathcal{E}_{\text{gen}}(g_\theta^{l_{\text{train}}}; l_{\text{test}}) \leq C_1^{(l_{\text{test}})} \cdot \mathcal{L}_{\text{misalign}}(g_\theta^{l_{\text{train}}}) + C_2^{(l_{\text{test}})} \cdot \mathcal{L}_{\text{train}}(g_\theta^{l_{\text{train}}}) + C_0^{(l_{\text{test}})}, \tag{28}$$

*where $C_i^{(l_{\text{test}})}$ are constants related to $l_{\text{test}}$. Specifically, the ratio $C_1^{(l_{\text{test}})}/C_2^{(l_{\text{test}})}$ becomes larger as $l_{\text{test}}$ increase. This indicates that as the testing length increases, the alignment loss becomes increasingly significant.*

*Proof.* We have:

$$\left\| g_\theta^{l_{\text{train}}}(\mathbf{x}_{[-l_{\text{test}}:]}) - y(\mathbf{x}_{[-l_{\text{test}}:]}) \right\|_2^2 \leq \left\| g_\theta^{l_{\text{train}}}(\mathbf{x}_{[-l_{\text{test}}:]}) - g_\theta^{l_{\text{train}}}(\mathbf{x}_{[-l:]}) \right\|_2^2$$
$$+ \left\| g_\theta^{l_{\text{train}}}(\mathbf{x}_{[-l:]}) - y(\mathbf{x}_{[-l_{\text{test}}:]}) \right\|_2^2,$$

where $l \in [l_{\text{train}}/2, l_{\text{train}}]$ is an arbitrary integer. Expanding the first term:

$$\left\| g_\theta^{l_{\text{train}}}(\mathbf{x}_{[-l_{\text{test}}:]}) - g_\theta^{l_{\text{train}}}(\mathbf{x}_{[-l:]}) \right\|_2^2$$

$$= \left\| \frac{1}{l_{\text{test}}} \mathbf{x}_0 W^Q (W^K)^\top \mathbf{x}_{[-l_{\text{test}}:]}^\top \mathbf{x}_{[-l_{\text{test}}:]} W^V W - \frac{1}{l} \mathbf{x}_0 W^Q (W^K)^\top \mathbf{x}_{[-l:]}^\top \mathbf{x}_{[-l:]} W^V W \right\|_2^2$$

$$= \left\| \mathbf{x}_0 W^Q (W^K)^\top \left( \frac{\mathbf{x}_{[-l_{\text{test}}:]}^\top \mathbf{x}_{[-l_{\text{test}}:]}}{l_{\text{test}}} - \frac{\mathbf{x}_{[-l:]}^\top \mathbf{x}_{[-l:]}}{l} \right) W^V W \right\|_2^2$$

$$= \left\| \mathbf{x}_0 W^Q (W^K)^\top \left( \frac{\mathbf{x}_{[-l_{\text{test}}:-l-1]}^\top \mathbf{x}_{[-l_{\text{test}}:-l-1]}}{l_{\text{test}}} - \frac{l_{\text{test}} - l}{l \cdot l_{\text{test}}} \mathbf{x}_{[-l:]}^\top \mathbf{x}_{[-l:]} \right) W^V W \right\|_2^2$$

$$\leq \left\| \mathbf{x}_0 W^Q (W^K)^\top \frac{\mathbf{x}_{[-l_{\text{test}}:-l-1]}^\top \mathbf{x}_{[-l_{\text{test}}:-l-1]}}{l_{\text{test}}} W^V W \right\|_2^2$$

$$+ \left\| \mathbf{x}_0 W^Q (W^K)^\top \left( \frac{l_{\text{test}} - l}{l \cdot l_{\text{test}}} \mathbf{x}_{[-l:]}^\top \mathbf{x}_{[-l:]} \right) W^V W \right\|_2^2.$$

For the first term, we have:

$$\left\| \mathbf{x}_0 W^Q (W^K)^\top \frac{\mathbf{x}_{[-l_{\text{test}}:-l-1]}^\top \mathbf{x}_{[-l_{\text{test}}:-l-1]}}{l_{\text{test}}} W^V W \right\|_2^2 \leq \frac{C_0 \|\mathbf{x}_0\|_2^2}{l_{\text{test}}^2} \left\| \mathbf{x}_{[-l_{\text{test}}:-l-1]}^\top \mathbf{x}_{[-l_{\text{test}}:-l-1]} \right\|_2^2$$

$$\leq \frac{C_0 (l_{\text{test}} - l)^2}{l_{\text{test}}^2},$$

where $C_0 = \left\| W^Q (W^K)^\top W^V W \right\|_2^2$ is a constant. We assume that $k = l_{\text{test}} - l$. For the second term, we have:

$$\left\| \mathbf{x}_0 W^Q (W^K)^\top \left( \frac{l_{\text{test}} - l}{l \cdot l_{\text{test}}} \mathbf{x}_{[-l:]}^\top \mathbf{x}_{[-l:]} \right) W^V W \right\|_2^2$$

$$= \frac{k^2}{l_{\text{test}}^2} \left\| \frac{1}{l} \mathbf{x}_0 W^Q (W^K)^\top \mathbf{x}_{[-l:]}^\top \mathbf{x}_{[-l:]} W^V W \right\|_2^2$$

$$\leq \frac{k^2}{l_{\text{test}}^2} \left\| \mathbf{x}_0 W^Q (W^K)^\top \left( \frac{\mathbf{x}_{[-l:]}^\top \mathbf{x}_{[-l:]}}{l} - \frac{\mathbf{x}_{[-(l-l'):]}^\top \mathbf{x}_{[-(l-l'):]}}{l - l'} \right) W^V W \right\|_2^2$$

$$+ \frac{k^2}{l_{\text{test}}^2} \left\| \frac{1}{l - l'} \mathbf{x}_0 W^Q (W^K)^\top \mathbf{x}_{[-(l-l'):]}^\top \mathbf{x}_{[-(l-l'):]} W^V W \right\|_2^2$$

$$= \frac{k^2}{l_{\text{test}}^2} \left\| g_\theta^{l_{\text{train}}} (\mathbf{x}_{[-l:]}) - g_\theta^{l_{\text{train}}} (\mathbf{x}_{[-(l-l'):]}) \right\|_2^2$$

$$+ \frac{k^2}{l_{\text{test}}^2} \left\| g_\theta^{l_{\text{train}}} (\mathbf{x}_{[-(l-l'):]}) \right\|_2^2$$

$$\leq \frac{k^2}{l_{\text{test}}^2} \left\| g_\theta^{l_{\text{train}}} (\mathbf{x}_{[-l:]}) - g_\theta^{l_{\text{train}}} (\mathbf{x}_{[-(l-l'):]}) \right\|_2^2$$

$$+ \frac{k^2}{l_{\text{test}}^2} \left\| g_\theta^{l_{\text{train}}} (\mathbf{x}_{[-(l-l'):]}) - y(\mathbf{x}_{[-(l-l'):]}) \right\|_2^2 + \frac{k^2}{l_{\text{test}}^2},$$

where $l' \geq 0$ is an arbitrary integer satisfying $l - l' \geq l_{\text{train}}/2$. Note that $k \leq l_{\text{test}} - l_{\text{train}}/2 := L$. Combining these results we have:

$$\left\| g_\theta^{l_{\text{train}}} (\mathbf{x}_{[-\mathbf{x}_{[-l_{\text{test}}:]}:]}) - y(\mathbf{x}_{[-l_{\text{test}}:]}) \right\|_2^2$$

$$\leq \frac{L^2}{l_{\text{test}}^2} \left\| g_\theta^{l_{\text{train}}} (\mathbf{x}_{[-l:]}) - g_\theta^l (\mathbf{x}_{[-(l-l'):]}) \right\|_2^2$$

$$+ \frac{L^2}{l_{\text{test}}^2} \left\| g_\theta^{l_{\text{train}}} (\mathbf{x}_{[-(l-l'):]}) - y(\mathbf{x}_{[-(l-l'):]}) \right\|_2^2$$

$$+ \left\| g_\theta^{l_{\text{train}}} (\mathbf{x}_{[-l:]}) - y(\mathbf{x}_{[-l:]}) \right\|_2^2$$

$$+ (1 + C_0) \frac{L^2}{l_{\text{test}}^2}.$$

We assume that the sampling process of $l$ and $l'$ is the same as the training sampling process. Therefore, taking the expectation over $\mathbf{x}, l, l'$ on both sides of the above equation gives:

$$\mathcal{E}_{\text{gen}} (g_\theta^{l_{\text{train}}}; l_{\text{test}}) = \mathbb{E}_{\mathbf{x}} \left\| g_\theta^{l_{\text{train}}} (\mathbf{x}_{[-l_{\text{test}}:]}) - y(\mathbf{x}_{[-l_{\text{test}}:]}) \right\|_2^2$$

$$\leq C_1^{(l_{\text{test}})} \cdot \mathcal{L}_{\text{misalign}} (g_\theta^{l_{\text{train}}}) + C_2^{(l_{\text{test}})} \cdot \mathcal{L}_{\text{train}} (g_\theta^{l_{\text{train}}}) + C_0^{(l_{\text{test}})}, \quad (29)$$

where $C_1^{(l_{\text{test}})} = L^2/l_{\text{test}}^2, C_2^{(l_{\text{test}})} = L^2/l_{\text{test}}^2 + 1, C_0^{(l_{\text{test}})} = (1 + C_0) L^2/l_{\text{test}}^2$. It's obvious that as $l_{\text{test}}$ increases, the ratio $L/l_{\text{test}}$ becomes larger, so the ratio $C_1^{(l_{\text{test}})}/C_2^{(l_{\text{test}})}$ also becomes larger. Therefore, as the testing length increases, the alignment loss becomes increasingly significant. $\square$

## C  ANALYSIS ON THE SUM PREDICTION TASK

We also examine the sum prediction task, where the label corresponds to the sum of the sequence. Similar to the length prediction task, the output space shifts as the sequence length increases. As a result, models struggle with length generalization in this task, as shown in Figure 4a. However, by applying the reparameterization technique proposed in Section 3, we observe a significant improvement in length generalization, as shown in Figure 4b. These results demonstrate the importance of output alignment in length generalization.

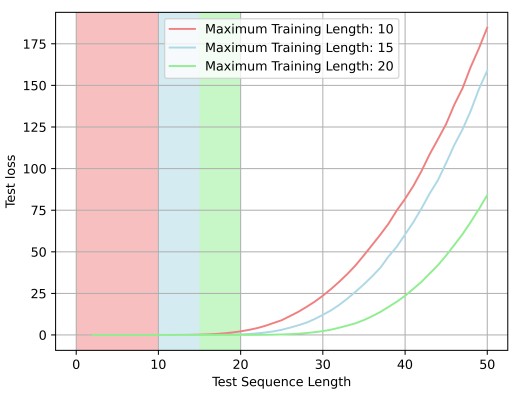

(a) Length generalization in sum prediction task with different maximum training sequence lengths

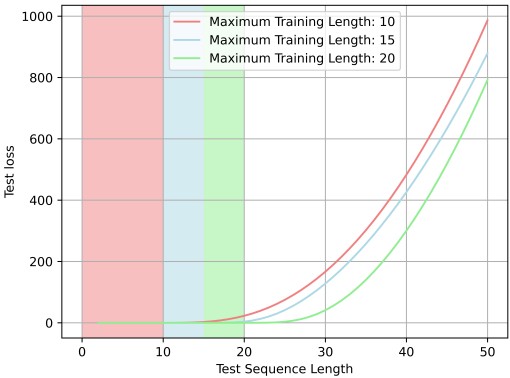

(b) Length generalization in length prediction task with different maximum training sequence lengths

Figure 3: Length generalization performance in the sum prediction and length prediction task with different maximum training sequence lengths. Although increasing the training length helps reduce the generalization error, the overall trend of increasing test loss remains unchanged.

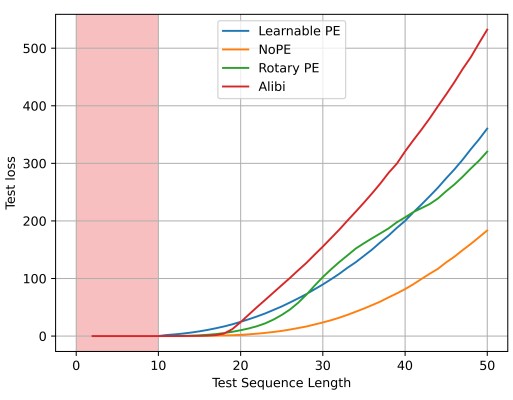

(a) Length generalization in the sum prediction task

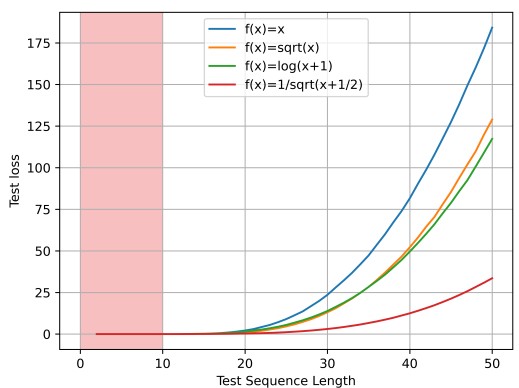

(b) Reparameterization in the sum prediction task

Figure 4: Length generalization in the sum prediction task. Explicit alignment of output space boosts length generalization performance.

## D    PYTORCH-LIKE CODE FOR IMPLEMENTATION OF $\mathcal{L}_{\text{train}}^*$

```
# An efficient implementation for the total training loss.
import torch
import random

def SCE(output1_prob, output2_prob):
    loss = torch.mean((torch.sum(- output1_prob * torch.exp(output2_prob)
        - output1_prob * torch.exp(output1_prob), -1)))

extra_len = random.randint(1, max_len//2)
data = get_data(seq_len=max_len+extra_len)

output1 = model(data[:, :max_len])
output2 = model(data[:, -max_len:])
prob1 = torch.nn.functional.log_softmax(output1.logits)
prob2 = torch.nn.functional.log_softmax(output2.logits)

# Select the overlapped part to calculate the misalign loss
prob1 = prob1[:, max_len//2+extra_len:]
prob2 = prob2[:, max_len//2:max_len-extra_len]
```

Table 9: Performance of the fine-tuned models using only cross-entropy loss (baseline) and an additional long-short misalignment loss on long-context modeling benchmark, LongBench-E score (Bai et al., 2023b) and perplexity on the 8k-length validation set. The fine-tuning sequence length is 4k, exactly the same as the training sequence length. We adopt two datasets: RedPajama-Book (Computer, 2023) and PG19 (Rae et al., 2019). The models finetuned with our proposed loss outperform the baseline across different model adaption strategies. The comparison is based on fixed total computation time. $(a/b)$ in the training steps mean $a$ steps for the baseline and $b$ steps for our method.

| Benchmark | LongBench-E ($\uparrow$) | | | Perplexity ($\downarrow$) | | |
|---|---|---|---|---|---|---|
| Training steps | 50/47 | 100/95 | 200/190 | 50/47 | 100/95 | 200/190 |
| *RedPajama-Book* | | | | | | |
| $\mathcal{L}_{\text{train}}$ (Baseline) | 22.7 | 23.8 | 24.7 | 7.21 | 6.56 | 6.12 |
| $\mathcal{L}_{\text{train}} + 0.1\mathcal{L}_{\text{misalign}}$ (Ours) | **23.1** | **25.1** | **26.4** | **6.92** | **6.27** | **5.91** |
| $\mathcal{L}_{\text{train}} + 0.5\mathcal{L}_{\text{misalign}}$ (Ours) | 21.7 | 23.4 | 24.5 | 7.62 | 7.16 | 6.61 |
| *PG19* | | | | | | |
| $\mathcal{L}_{\text{train}}$ (Baseline) | 20.2 | 21.4 | 22.5 | **8.92** | **7.89** | 7.45 |
| $\mathcal{L}_{\text{train}} + 0.1\mathcal{L}_{\text{misalign}}$ (Ours) | **20.7** | **22.0** | **25.1** | 9.02 | 8.01 | **7.39** |
| $\mathcal{L}_{\text{train}} + 0.5\mathcal{L}_{\text{misalign}}$ (Ours) | 19.6 | **22.0** | 23.3 | 9.82 | 8.62 | 8.29 |

```
19  loss_ce = (output1.loss + output2.loss) / 2
20  loss_misalign = SCE(prob1, prob2)
21
22
23  loss_total = loss_ce + alpha * loss_misalign
```

# E  COMPARISON UNDER SAME COMPUTATION TIME

To account for the additional computation time required to calculate $\mathcal{L}_{\text{misalign}}$ we will compare our methods with the baseline as in Table 2 and Table 3 based on total computation time. we compare our method with the baseline under equivalent total computation costs. Our method introduces an additional computational overhead of approximately 3% to 5% per step. To ensure fairness, we adjust the number of training steps proportionally. For example, when the baseline is trained for 50, 100, and 200 steps, our method is trained for 47, 95, and 190 steps, respectively, achieving comparable total computation times. We observe that the performance trends remain consistent: our method continues to outperform the baseline under equivalent computation time. This underscores the efficiency of our approach despite the minor additional cost.

# F  ADDITIONAL EXPERIMENT ON MEAN PREDICTION TASK WITH A DIFFERENT DATASET SETTING

In the synthetic experiments in Section 3, 0 and 1 have an equal probability. The mean value of 50 such samples will nearly obey the normal distribution $\mathcal{N}(0.5, 0.005)$. This means predicting 0.5 under a length of 50 would yield an approximate test loss of 0.005. However, we would like to clarify that the test loss of NoPE remains around 1e-5 (indicated by the orange line in Figure 1(a)), which is two orders of magnitude smaller than 0.005. This indicates that the model predicts the mean values of the sequences with high precision, rather than simply guessing a fixed number. Therefore, the conclusion that the model can have good length generalization ability in the mean prediction task is reasonable.

Besides, in order to avoid a trivial solution of predicting 0.5, we conduct an additional experiment on the mean prediction task. To build the sample of length $l$, we first sample the number of 1 uniformly from $[0, l]$ and then randomly build the sequence. In this way, the mean value of the sequence uniformly spans from 0 to 1, avoiding trivial prediction. The experimental results are shown in

Table 10: Performance of the finetuned models using only cross-entropy loss (baseline) and an additional long-short misalignment loss on long-context modeling benchmark, LongBench-E score (Bai et al., 2023b) and perplexity on the 8k-length validation set. The fine-tuning sequence length is 8k. We adopt two kinds of model adjustments: LongQLora (Yang, 2023) and EABF (Zhang et al., 2024b). The models finetuned with our proposed loss outperform the baseline across different model adaption strategies. The comparison is based on fixed total computation time. $(a/b)$ in the training steps mean $a$ steps for the baseline and $b$ steps for our method.

| Benchmark | LongBench-E ($\uparrow$) | | | Perplexity ($\downarrow$) | | |
| Training steps | 50/47 | 100/95 | 200/190 | 50/47 | 100/95 | 200/190 |
|---|---|---|---|---|---|---|
| *LongQLora* | | | | | | |
| $\mathcal{L}_{\text{train}}$ (Baseline) | **21.9** | 22.1 | 23.4 | 6.82 | **6.41** | 5.82 |
| $\mathcal{L}_{\text{train}} + 0.1\mathcal{L}_{\text{misalign}}$ (Ours) | 21.6 | 23.1 | **25.7** | **6.79** | 6.42 | **5.74** |
| $\mathcal{L}_{\text{train}} + 0.5\mathcal{L}_{\text{misalign}}$ (Ours) | 21.1 | **23.6** | 24.9 | 7.19 | 6.65 | 5.96 |
| *EABF* | | | | | | |
| $\mathcal{L}_{\text{train}}$ (Baseline) | 22.1 | 22.9 | 23.6 | **6.89** | 6.52 | 6.01 |
| $\mathcal{L}_{\text{train}} + 0.1\mathcal{L}_{\text{misalign}}$ (Ours) | **23.0** | **23.6** | **24.5** | 7.01 | **6.48** | **5.86** |
| $\mathcal{L}_{\text{train}} + 0.5\mathcal{L}_{\text{misalign}}$ (Ours) | 22.2 | 23.1 | 23.8 | 7.32 | 6.88 | 6.42 |

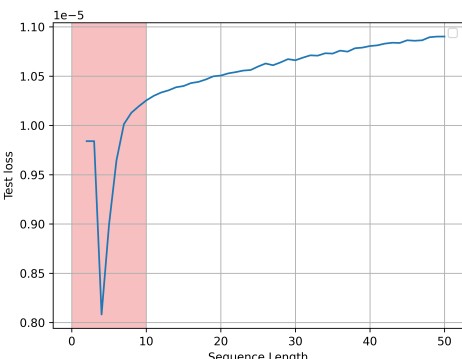

Figure 5: Length generalization in the mean prediction task with a different dataset setting. To build the sample of length $l$, we first sample the number of 1 uniformly from $[0, l]$ and then randomly build the sequence. In this way, the mean value of the sequence uniformly spans from 0 to 1, avoiding trivial prediction. The model can still achieve good length generalization, which is consistent with our previous results.

Figure 5, where the model can still achieve good length generalization, consistent with our previous results.

## G COMPARISON WITHOUT USING MODEL ADJUSTMENT METHODS

In order to compare our proposed method directly with other methods, we conduct experiments under the following settings: (1) Raw fine-tuning: The baseline without any additional techniques; (2) Only LongQLora: Incorporating the LongQLora (Yang, 2023) method as described in Table 3; (3) Only EABF: Incorporating the EABF method as described in Table 3 (Chen et al., 2023a); (4) Only our proposed output alignment technique: Applying our method without other enhancements. The results are shown in Table 11. From the results, we observe: (1) Our method alone outperforms raw fine-tuning, demonstrating its effectiveness in improving length generalization; (2) Positional encoding-based approaches, such as LongQLora and EABF, achieve higher performance compared to using our method alone. When combined with positional encoding-based approaches, our method consistently yields additional improvements, as shown in Table 2 and Table 3 in the paper. It is important to note that our proposed output alignment technique operates from the perspective of loss

Table 11: Comparison between our proposed method directly with other methods without using model adjustment methods.

| Benchmark | LongBench-E (↑) | | | Perplexity (↓) | | |
|---|---|---|---|---|---|---|
| Training steps | 50 | 100 | 200 | 50 | 100 | 200 |
| (1) Raw finetuning | 14.9 | 16.1 | 18.4 | 21.73 | 18.52 | 16.13 |
| (2) Only with LongQLora | 21.9 | 22.1 | 23.4 | 6.82 | 6.41 | 5.82 |
| (3) Only with EABF | 21.4 | 22.7 | 24.5 | 6.89 | 6.52 | 6.01 |
| (4) Only with output alignment technique ($\alpha = 0.1$) | 15.4 | 18.2 | 20.9 | 19.25 | 15.47 | 11.24 |

design, focusing on the model's output alignment during training. In contrast, positional encoding-based approaches primarily address the input representation. **These two strategies are orthogonal and can be seamlessly integrated.** The experimental results demonstrate that incorporating our method with positional encoding techniques enhances long-context modeling capabilities beyond what is achieved by either approach alone.

