# OpenReview forum: "Output Alignment: A Top-down Approach to Length Generalization"
_ICLR.cc/2025/Conference — Submitted to ICLR 2025_

### Official Review · Reviewer_NSDT · 2024-11-03

**Soundness:** 2
**Presentation:** 4
**Contribution:** 3
**Rating:** 5
**Confidence:** 3

**Summary:**

This paper presents output alignment as a method for enhancing the length generalization performance. The author initiates with the observation that the output space is significant in length generalization as demonstrated by a synthetic task. Through both theoretical and experimental analyses, the author validates that the output space has an impact on performance. The paper then proceeds to propose an approach for constructing a robust output space for natural language tasks. The performance of the proposed loss is advantaged by the experiments.

**Strengths:**

The strength is evident:
1. A novel perspective from the output space of the Language Learning Model (LLM) when addressing the length generalization problem rather than the common input space analysis.
2. A well-defined synthetic task is proposed to drive the motivation and is strongly supported by theoretical analysis and experiments.
3. The proposed training loss is straightforward to implement and brings a relatively significant improvement to the length generalization performance.

**Weaknesses:**

The weaknesses are as follows:

1. There is a lack of understanding regarding the design of mean value prediction tasks. Given that 0 and 1 have an equal probability (as indicated on line 143), does this imply that both 0 and 1 have a probability of 0.5? If so, does it mean that predicting 0.5 under a length of 50 would yield an approximate test loss of 0.005? Despite the actual test loss being around 0.001, this design appears suboptimal.

2. The distinct scaling behavior in mean and length is highly evident, particularly when a feed-forward layer is added to predict a scalar of the length. Predicting out-of-distribution values is a well-known challenge. Since the input and weight have fixed scales, but the output is expected to be scalable with respect to the length, the synthetic tasks seem rather explicit. This is not necessarily a weakness but rather emphasizes the need to clarify the purpose of such a preliminary experiment, which will be my next question.

3. The question arises as to how the synthetic task is actually related to natural language cases where the misalign loss is proposed. The only connection observed is “output space matter”. However, there appears to be a fundamental difference in “the output space/output distribution/output alignment” between the two scenarios. In the former case, it is more of a “scale/magnitude” issue. As the scale of x is changed to logx or 1/sqrt(x), the scale/magnitude becomes less out-of-distribution when scaling the length. In contrast, in the natural language case, it is primarily an “actual distribution” issue since the “scale/magnitude” does not change significantly. While both cases can be understood, the connection between them remains unclear. From this perspective, the term “misalignment” seems to be misused. The author could provide further clarification.

4. The experiment appears to be relatively weak. For natural language tasks, the author only evaluates it on LongBench, and the performance improvement is marginal. There are two potential ways to enhance the robustness of the results. Firstly, conducting experiments on more tasks such as Need-in-a-haystack, InfiniteBench, BABILong, etc. Secondly, analyzing the generations of the baselines and the proposed methods and providing randomly sampled output samples to show that the use of the additional loss is demonstrably better.

**Questions:**

Small issues in writing:

1. L150,  ... embedding used, the test loss dramatically...  => ... embedding used, the test loss of length prediction task dramatically...
2. L298-L299, A bit confusing about sufficies.

---

> ### Author Response · Authors · 2024-11-23
> **Reply to Reviewer NSDT (1/3)**
>
> We thank Reviewer NSDT for careful reading, detailed comments and appreciation of the novelty and the significance of our work. Below, we will address your concerns in the following points.
>
> ---
>
> Q1. There is a lack of understanding regarding the design of mean value prediction tasks. Given that 0 and 1 have an equal probability (as indicated on line 143), does this imply that both 0 and 1 have a probability of 0.5? If so, does it mean that predicting 0.5 under a length of 50 would yield an approximate test loss of 0.005? Despite the actual test loss being around 0.001, this design appears suboptimal.
>
> A1.  Indeed, your calculation is right. In our experiments, 0 and 1 have an equal probability. The mean value of 50 such samples will nearly obey the normal distribution $\mathcal{N}(0.5, 0.005)$. This means predicting 0.5 under a length of 50 would yield an approximate test loss of 0.005. However, we would like to clarify that the test loss of NoPE keeps around 1e-5 (indicated by the orange line in Figure 1(a)), which is two orders of magnitude smaller than 0.005. This indicates that the model predicts the mean values of the sequences with high precision, rather than simply guessing a fixed number. Therefore, the conclusion that model can have good length generalization ability in the mean prediction task is reasonable.
>
> Besides, in order to avoid a trivial solution of predicting 0.5, we conduct an additional experiment on the mean prediction task. To build the sample of length $l$, we first sample the number of 1 uniformly from $[0, l]$ and then randomly build the sequence. In this way, the mean value of the sequence uniformly spans from 0 to 1, avoiding trivial prediction. The experimental results are shown in Figure 5 in Appendix F, where the model can still achieve good length generalization, consistent with our previous results.
>
> ---
>
> Q2. Since the input and weight have fixed scales, but the output is expected to be scalable with respect to the length, the synthetic tasks seem rather explicit. This is not necessarily a weakness but rather emphasizes the need to clarify the purpose of such a preliminary experiment.
>
> A2.
>
> Thank you for your thoughtful feedback. To clarify, the synthetic tasks in Section 3 are designed to provide a controlled setting to analyze the challenges of length generalization and the importance of output alignment. While the inputs and weights indeed have fixed scales, it is not inherently trivial that the output would fail to generalize across varying lengths. For example, architectures such as GRUs and LSTMs are capable of generating outputs with scales that differ from their inputs under certain conditions [1]. This highlights that neural networks, in general, can adapt to scale differences, making the Transformer’s difficulty with the length prediction task a noteworthy observation.
>
> The purpose of these experiments is **to motivate the need for our proposed output alignment techniques**. By comparing the mean prediction task (where the output is aligned) and the length prediction task (where it is not), we provide an intuitive and illustrative example of how output alignment affects generalization. The great effectiveness of our reparameterization technique further emphasizes this point.
>
> Based on your suggestion, we have revised Section 3 to focus more clearly on these motivations. The section has been condensed to slightly more than one page, with detailed theoretical content and explanations moved to the appendix. This refinement aims to make the purpose of the synthetic tasks clearer. We hope this revision addresses your concern and enhances the clarity and fluency of the manuscript.
>
> [1] Mirac Suzgun et al, LSTM Networks Can Perform Dynamic Counting, Proceedings of the Workshop on Deep Learning and Formal Languages, 2019.

---

> ### Author Response · Authors · 2024-11-23
> **Reply to Reviewer NSDT (2/3)**
>
> Q3. The question arises as to how the synthetic task is actually related to natural language cases where the misalign loss is proposed. In the synthetic task case, it is more of a “scale” issue. In the natural language case, it is an “actual distribution” issue. From this perspective, the term “misalignment” seems to be misused. The author could provide further clarification.
>
> A3. To address your concern, we clarify that we view the scale difference in synthetic tasks as a specific case of distributional variation. For example, in the length prediction task, the output distribution of input sequences with length $l$ is just one point distribution $O(l)$. In the sum prediction task (illustrated in Appendix C), the output distribution of input sequences with length $l$ is the binomial distribution $B(l, 0.5)$. These examples illustrate that the output distributions vary significantly as $l$ increases. Through our proposed reparameterization technique, these variations are explicitly aligned across different lengths.
>
> To further clarify the relationship between synthetic tasks and natural language tasks, we summarize key differences and similarities in the following table:
>
> |  | Synthetic Tasks | Language Tasks |
> | --- | --- | --- |
> | Output Space | $\mathbb{R}$ | $L_1$ unit ball in $\mathbb{R}^{|\mathcal{V}|}$ |
> | Specific Task | Length/Sum prediction | Next token prediction |
> | Output Distribution Variation across Lengths | Exist | Exist |
> | Priori on Output Distribution | Explicit and predefined | Implicit and task-dependent |
> | Alignment Technique | Explicit reparameterization | Regularization loss across lengths |
> | Does alignment technique decrease output distribution variation? | Yes (explicitly) | Yes (implicitly through optimization) |
> | Does alignment technique improve length generalization? | Yes | Yes |
>
> From this perspective, the term “misalignment” is applicable to both scenarios, as **it consistently describes the lack of alignment between output distributions across different input sequence lengths**. While synthetic tasks exhibit more explicit and measurable variations (e.g., scale), natural language tasks involve subtler, often implicit shifts in distribution
>
> We have added a discussion in Section 6 of the manuscript to explicitly discuss the relationship between these two types of issues and highlight how they can be analyzed within a unified framework. We hope this clarification and the revision address your concerns.

---

> ### Author Response · Authors · 2024-11-23
> **Reply to Reviewer NSDT (3/3)**
>
> Q4. The authors could (1) conduct experiments on more tasks such as Need-in-a-haystack, InfiniteBench, BABILong, (2) analyze the generations of the baselines and the proposed methods and provide randomly sampled output samples to show that the use of the additional loss is demonstrably better.
>
> A4. Thanks for your suggestions! We conduct evaluation experiments on BABILong, which comprises question-answering tasks where the supporting facts for each question are situated at specific positions within the context. The results can be viewed in Table 6 and Table 7. We also copy these results in the following tables:
>
> - Evaluation is performed on input sequences of lengths 4K, 8K, and 16K, with the overall results summarized in Table 6.
>
> | Evaluation Length | 4K | 8K | 16K |
> | --- | --- | --- | --- |
> | $L_{train}$ (Baseline) | 48.2 | 42.4 | 37.9 |
> | $L_{traiin}+0.1L_{misalign}$ | **49.1** | **44.4** | **40.1** |
>
> The results indicate that our proposed method consistently outperforms the baseline across all evaluated lengths. Specifically, our method achieves a performance gain of $2.0\%$ at length 8K and $2.2\%$ at length 16K. These results demonstrate the effectiveness of our regularization loss in enhancing length generalization.
>
> - Additionally, we analyze the impact of the supporting fact's position within the input context using the QA1 task from BABILong, where each question is associated with a single supporting fact. Results from this analysis are presented in Table 7.
>
> |  | Fact Depth=0% | Fact Depth=25% | Fact Depth=50% | Fact Depth=75% |
> | --- | --- | --- | --- | --- |
> | $L_{train}$ (Baseline) | 75 | 64 | 30 | 69 |
> | $L_{train}+0.1L_{misalign}$ | 73 | 64 | 38 | 74 |
>
> These results offer two key insights: (1) **Performance with early-context facts:** When the supporting fact is located at the beginning of the input context (fact depth = 0), our method achieves performance comparable to the baseline. This suggests that, despite the form of the regularization potentially encouraging the model to neglect earlier contexts, it does not lead to this behavior in practice. (2) **Performance with middle-context facts:** When the supporting fact is positioned in the middle of the context (fact depth = 50 or 75), our method shows considerable improvement over the baseline. This indicates that our approach effectively mitigates the "loss-in-the-middle" phenomenon [2], a common challenge in large language models.
>
> The full setting and results can be viewed in Section 5.4 in the revision. We hope these results can ease your concern on the effectiveness of the proposed method.
>
> [2] Nelson F Liu et al, Lost in the middle: How language models use long contexts. Transactions of the Association for Computational Linguistics, 2024.
>
> ---
>
> Q5. Small issues in writing.
>
> A5. Thanks for pointing them out. We have revised them in the manuscript accordingly.
>
> ---
>
> Thanks again for your careful reading and detailed review. Hope our explanations and extended empirical justifications could address your concern. Please let us know if you have additional questions.

---

> ### Author Response · Authors · 2024-11-25
> **Your invaluable input is needed.**
>
> Dear Reviewer NSDT,
>
> Thank you for your thoughtful feedback on our manuscript. We have carefully addressed each of your questions in our detailed response. We would greatly appreciate it if you could review our revisions and let us know if they address your concerns.
>
> We are encouraged by Reviewer AF82’s acknowledgment of our revisions, which they found satisfactory, and their decision to raise the score from 3 to 6. Your further input would be invaluable to us as we continue to refine our work.
>
> Thank you again for your time and effort, and we hope you have a wonderful day!
>
> Best regards,
>
> Authors

---

> ### Author Response · Authors · 2024-12-02
> **Your further inputs are greatly appreciated. Only one day left.**
>
> Dear Reviewer NSDT,
>
> For your raised questions, we prepared a detailed response to address your concerns. We were hoping to hear your feedback on them.
>
> As there is only one day left for the reviewer and author discussions and we understand that everyone has a tight schedule, we kindly wanted to send a gentle reminder to ensure that our response sufficiently addressed your concerns or if there are further aspects we need to clarify.
>
> If you could find the time to provide your thoughts on our response, we would greatly appreciate it.
>
> Best, Authors

---

### Official Review · Reviewer_AF82 · 2024-11-03

**Soundness:** 3
**Presentation:** 3
**Contribution:** 3
**Rating:** 6
**Confidence:** 3

**Summary:**

The paper addresses the challenge of length generalization in LLMs. Instead of focusing on input structure or positional encoding, the authors present a novel approach centered on the output distribution to enhance length generalization. They highlight the concept of output alignment and introduce the Long-Short Misalignment metric to measure and quantify the consistency of output distributions for varying sequence lengths. Empirical results from synthetic tasks and natural language tasks validate their approach, demonstrating that output misalignment significantly impacts model performance on longer sequences. The paper further proposes a regularization loss based on their metric, which improves length generalization as shown through extensive experiments.

**Strengths:**

1. The paper is well-written and easy to follow. The explanations and the illustrations are clear.
2. The proposed output alignment method for improving LLMs' length generalization is novel. The long-short misalignment regularization loss can be beneficial for future long-context LLM training.
3. The long-short misalignment loss is an interesting and useful probe for long-context LLM performance.

**Weaknesses:**

1. Although the authors include extensive theoretical analysis of the proposed metric and method, the proofs of the theorems are based on some very strong assumptions, which need to be justified. More details in the Questions part.
2. The analysis of the synthetic tasks and the natural language tasks seems disjointed. The cause of the errors is different for these two types of tasks. This makes the OutRep method proposed in Section 3.2 seem unrelated to the rest of the paper and has a very limited use case on only non-pretrained small models. It would be beneficial to make a stronger connection between Section 3.2 and the following sections.
3. The experiments show that a higher regularization coefficient can negatively affect performance. More analysis is needed to mitigate the risk of over-regularization.

**Questions:**

1. The proof of Theorem 1 is based on an autoregressive expression (Eqn 9). However, it is unclear why self-attention can be reformulated in this way. Also, it seems like Assumption 1 assumes that the function $K$ can disregard part of the input context. In an extreme situation where $l_{train}$ is very small, this assumption means that $K$ neglects a large portion of the prompt. Could you please elaborate more on the validity of Eqn 9 and Assumption 1?
2. Assumption 2 assumes that a Transformer model is Lipschitz continuous. However, this is not true for Transformers. Please refer to [1].
3. Regarding the long-short misalignment regularization, in a situation where $l_{train}$ is small, will the model learn to neglect some proportion of the input at the front?
4. Some key references are missing. For example, the mean prediction task is similar to the parity counter task in [2], and there has been similar length extrapolation analysis on synthetic tasks in [3].
5. In Line 400, the sequences should overlap by $l_{extra}$ tokens.

[1] Hyunjik Kim, George Papamakarios, and Andriy Mnih. 2021. The Lipschitz Constant of Self-Attention. ICML.

[2] Bingbin Liu, Jordan T Ash, Surbhi Goel, Akshay Krishnamurthy, and Cyril Zhang. 2022. Transformers Learn Shortcuts to Automata. ICLR.

[3] Suyuchen Wang, Ivan Kobyzev, Peng Lu, Mehdi Rezagholizadeh, and Bang Liu. 2024. Resonance RoPE: Improving Context Length Generalization of Large Language Models. ACL Findings.

---

> ### Author Response · Authors · 2024-11-23
> **Reply to Reviewer AF82 (1/3)**
>
> We thank reviewer AF82 for careful reading, detailed comments and appreciation of the novelty and good presentation of our work. Below, we will address your concerns in the following points.
>
> ---
>
> Q1. The analysis of the synthetic tasks and the natural language tasks seems disjointed. It would be beneficial to make a stronger connection between Section 3.2 and the following sections.
>
> A1. Following your insightful suggestions, we refine Section 3 to slightly more than 1 page in the revision by removing the theoretical part and most of the explanation. Now, Section 3 only serves as a motivation for pointing out the critical role of output space in the length generalization, as the empirical results in the synthetic tasks are direct and significant.
>
> Subsequently, we add more explanation and empirical results in Section 4 and Section 5, shifting the main focus of the paper towards analyzing the phenomenon and proposing solutions in the context of natural language tasks. We hope that this revision can improve the clarity and cohesion of the paper. We also add a discussion between the synthetic tasks and the natural tasks in the new-added Section 6, with the following comparison table:
>
> |  | Synthetic Tasks | Language Tasks |
> | --- | --- | --- |
> | Output Space | $\mathbb{R}$ | $L_1$ unit ball in $\mathbb{R}^{\|\mathcal{V}\|}$ |
> | Specific Task | Length/Sum prediction | Next token prediction |
> | Output Distribution Variation across Lengths | Exist | Exist |
> | Priori on Output Distribution | Explicit and predefined | Implicit and task-dependent |
> | Alignment Technique | Explicit reparameterization | Regularization loss across lengths |
> | Does alignment technique decrease output distribution variation? | Yes (explicitly) | Yes (implicitly through optimization) |
> | Does alignment technique improve length generalization? | Yes | Yes |
>
> From this perspective, while these two types of tasks differ significantly in their specific forms and output space, they share a common challenge: **output distribution misalignment across different input lengths**. Our analysis highlights that employing an alignment technique--whether explicit reparameterization in synthetic tasks or regularization in natural language tasks--can effectively mitigate this misalignment. This mitigation directly enhances the model's length generalization ability, demonstrating that output distribution alignment will help improve length generalization. We hope this revision and discussion can ease your concerns.
>
> ---
>
> Q2. The experiments show that a higher regularization coefficient can negatively affect performance. More analysis is needed to mitigate the risk of over-regularization.
>
> A2. Sure. To address your concern, we conducted additional ablation experiments to analyze the impact of the regularization coefficient $\alpha$ on model performance. In addition to the settings already provided in the paper ($\alpha = 0$ as the baseline, $\alpha = 0.1$, and $\alpha = 0.5$), we evaluated $\alpha = 0.3$ and $\alpha = 1.0$ under the same experimental conditions as Table 2, using CLEX as the model adjustment. The updated results are summarized in Table 4 in the revision and shown below:
>
> | Benchmark |  | LongBench-E |  |  | Perplexity |  |
> | --- | --- | --- | --- | --- | --- | --- |
> | Training steps | 50 | 100 | 200 | 50 | 100 | 200 |
> | (1) $L_{train}$ (Baseline) | 22.7 | 23.8 | 24.7 | 7.21 | 6.56 | 6.12 |
> | (2) $L_{train}+0.1L_{misalign}$ | 23.1 | 25.2 | 26.6 | **6.89** | **6.24** | **5.88** |
> | (3) $L_{train}+0.3L_{misalign}$ | **23.4** | **25.8** | **27.1** | 6.95 | 6.35 | 5.98 |
> | (4) $L_{train}+0.5L_{misalign}$ | 21.9 | 23.7 | 24.7 | 7.44 | 7.01 | 6.54 |
> | (5) $L_{train}+1.0L_{misalign}$ | 18.2 | 19.4 | 19.9 | 16.21 | 14.12 | 12.92 |
>
> The results reveal the following trend:
>
> - Performance peaks for $\alpha$ values in the range [0.1, 0.3] in **both evaluation metrics**.
> - Larger values of $\alpha$ (e.g., $\alpha = 0.5$ or $\alpha = 1.0$) lead to a significant decline in performance, confirming the risks of over-regularization.
>
> These findings highlight the importance of selecting a moderate value for $\alpha$. We suggest using an coefficient $\alpha$ between 0.1 and 0.3 as default to mitigate the risk of over-regularization.

---

> ### Author Response · Authors · 2024-11-23
> **Reply to Reviewer AF82 (2/3)**
>
> Q3. The proof of Theorem 1 is based on an autoregressive expression (Eqn 9). However, it is unclear why self-attention can be reformulated in this way. Also, it seems like Assumption 1 assumes that the function $K$ can disregard part of the input context. In an extreme situation where $l_{train}$ is very small, this assumption means that $K$ neglects a large portion of the prompt. Could you please elaborate more on the validity of Eqn 9 and Assumption 1?
>
> A3.
>
> We agree that these assumptions may appear strong in their original form. In response, we have revised the theoretical framework to adopt a more moderate assumption. Specifically, we now assume the model incorporates a linear attention mechanism. This assumption is also adopted by prior theoretical studies on length generalization [1, 2].
>
> Under this revised framework, we re-derived the theoretical results and observed that our main conclusions remain consistent:
>
> - The model demonstrates strong length generalization ability in the **mean prediction task**.
> - However, it struggles to generalize effectively in the **sum prediction** and **length prediction tasks**.
>
> Due to the structural adjustment of our revised manuscript, Theorem 1 is swapped to Theorem 2. The full statement and proof of Theorem 2 (the original Theorem 1) can be viewed in Appendix B.1. We hope this revision adequately addresses your concerns and improves the robustness of our theoretical analysis.
>
> [1] Qi Zhang et al, Look ahead or look around? A theoretical comparison between autoregressive and masked pretraining. ICML, 2024.
>
> [2] Yue M. Lu et al, Asymptotic theory of in-context learning by linear attention. arXiv, 2405.11751.
>
> ---
>
> Q4. Assumption 2 assumes that a Transformer model is Lipschitz continuous. However, this is not true for Transformers.
>
> A4. We revised our theoretical framework as stated in our earlier response. Specifically, we now assume that the model incorporates a linear attention block. Under this revised framework, we derive the following generalization guarantee:
>
> $$\epsilon(g_{\theta}^{l_{train}};l_{test})\leq L^2/l^2_{test}\cdot L_{misalign}(g_{\theta}^{l_{train}})+(L^2/l^2_{test}+1)\cdot L_{train}(g_{\theta}^{l_{train}})+C,$$
>
> where:
>
> - $\epsilon$ is the generalization loss of model $g_{\theta}^{l_{train}}$ with train length $l_{train}$,
> - $l_{test}$ is the test length,
> - $L=l_{test}-l_{train}/2$,
> - $L_{misalign}$ is the long-short misalignment loss,
> - $L_{train}$ is the training loss, and
> - $C$ is a constant
>
> This generalization guarantee provides two key insights:
>
> - **Role of Misalignment Loss**: Optimizing $L_{misalign}$ is critical for improving length generalization.
> - **Impact of Test Length**: As the test length $l_{test}$ increases, the coefficient of $L_{misalign}$ also grows, indicating that minimizing $L_{misalign}$ becomes increasingly important for long-context scenarios.
>
> Due to the structural adjustment of our revised manuscript, Theorem 3 is swapped to Theorem 1. The full statement and proof of Theorem 1 (the original Theorem 3) can be viewed in Appendix B.2. We hope that this updated framework addresses the concerns about the assumption while maintaining the robustness of our theoretical conclusions.
>
> ---
>
> Q5. Regarding the long-short misalignment regularization, in a situation where $l_{train}$ is small, will the model learn to neglect some proportion of the input at the front?
>
> A5. To address your concern, we conduct experiments on the BABILong benchmark [3], a challenging reasoning-in-a-haystack task designed for long-context evaluation. BABILong consists of question-answering tasks where the supporting fact for each question is located at a specific position within the context. We analyze the impact of the supporting fact's position within the input context using the QA1 task from BABILong, where each question is associated with a single supporting fact. Results from this analysis are presented in Table 7 in the revision and below:
>
> |  | Fact Depth=0% | Fact Depth=25% | Fact Depth=50% | Fact Depth=75% |
> | --- | --- | --- | --- | --- |
> | $L_{train}$ (Baseline) | 75 | 64 | 30 | 69 |
> | $L_{train}+0.1L_{misalign}$ | 73 | 64 | 38 | 74 |
>
> When the supporting fact is located at the beginning of the input context (fact depth = 0), our method achieves performance comparable to the baseline (73 v.s. 75). These findings suggest that the proposed alignment technique does not cause the model to disregard information from the front of the input sequence, even when $l_{train}$ is relatively small.
>
> [3] Yuri Kuratov et al, Babilong: Testing the limits of llms with long context reasoning-in-a-haystack, arxiv 2406.10149.

---

> ### Author Response · Authors · 2024-11-23
> **Reply to Reviewer AF82 (3/3)**
>
> Q6. Some key references are missing. For example, the mean prediction task is similar to the parity counter task in [4], and there has been similar length extrapolation analysis on synthetic tasks in [5].
>
> A6. Thanks for pointing them out! [4] discovers that Transformer will learn shortcut through the study on various synthetic tasks. [5] proposes a novel approach designed to narrow the generalization gap by refining the interpolation of RoPE features for OOD positions and provides length extrapolation analysis on the feature gap. In the revision. we have added these discussions in the related work section.
>
> [4] Bingbin Liu, Jordan T Ash, Surbhi Goel, Akshay Krishnamurthy, and Cyril Zhang. Transformers Learn Shortcuts to Automata. ICLR, 2022.
>
> [5] Suyuchen Wang, Ivan Kobyzev, Peng Lu, Mehdi Rezagholizadeh, and Bang Liu. Resonance RoPE: Improving Context Length Generalization of Large Language Models. ACL Findings, 2024.
>
> ---
>
> Q7. In Line 400, the sequences should overlap by $l_{extra}$ tokens.
>
> A7. There may be some misunderstanding about the overlap between the two sequences. Let me clarify this. Given that the first sequence spans from $1$ to $l_{train}$ and the second sequence spans from $l_{extra}+1$ to $l_{extra}+l_{train}$. The overlap between the two sequences starts at token $l_{extra}+1$ and continues to token $l_{train}$, resulting in an overlap of $l_{train}-l_{extra}$ tokens, instead of $l_{extra}$ tokens. We add this clarification in the revised paper to help avoid such misunderstanding.
>
> ---
>
> Thanks again for your careful reading and detailed review. Hope our explanations and extended empirical justifications could address your concern. Please let us know if you have additional questions.

---

> ### Comment · Reviewer_AF82 · 2024-11-25
>
> Thank you for your thorough response and the additional efforts in revising the manuscript. I appreciate the clarifications and added results, which address many of my initial concerns. I have a couple of further questions regarding the newly included results and assumptions:
>
> 1. From Table 4, it appears that $\alpha=0.3$ yields the best performance for the downstream task LongBench-E. Could you clarify the rationale for selecting $\alpha=0.1$ in the BABILong experiments, given this observation?
>
> 2. Regarding the assumptions, I sincerely appreciate the extensive revision and the detailed explanations. As the experiments and qualitative analysis focus on vanilla Transformers, a brief discussion of how the assumption of linear attention could be extended or applied to vanilla Transformers would enhance the theoretical applicability and further strengthen the manuscript.
>
> Given the substantial revisions and the effort to address my previous concerns, I am inclined to raise my score. Thank you again for your dedication to improving the paper.

---

> > ### Author Response · Authors · 2024-11-25
> > **Furthur Reply to Reviewer AF82**
> >
> > Thank you for your positive feedback and for considering raising your score. We sincerely appreciate your detailed questions and valuable suggestions. Below, we address your two points in detail.
> >
> > ---
> >
> > Q1. From Table 4, it appears that $\alpha=0.3$ yields the best performance for the downstream task LongBench-E. Could you clarify the rationale for selecting $\alpha=0.1$ in the BABILong experiments, given this observation?
> >
> > A1. Thank you for raising this point. While $\alpha=0.3$ yields the best performance for the downstream task LongBench-E, $\alpha=0.1$ achieves the best perplexity on the 8k-length validation set, as shown in Table 4. Since both $\alpha=0.1$ and $\alpha=0.3$ perform well, we initially selected $\alpha=0.1$ for the BABILong experiments **without specific preference.**
> >
> > To further address your concern, we conducted additional experiments on BABILong with $\alpha=0.3$ under settings similar to Table 7. The results are presented below:
> >
> > |  | Fact Depth=0% | Fact Depth=25% | Fact Depth=50% | Fact Depth=75% |
> > | --- | --- | --- | --- | --- |
> > | $L_{train}$ (Baseline) | 75 | 64 | 30 | 69 |
> > | $L_{train}+0.1L_{misalign}$ | 73 | 64 | 38 | 74 |
> > | $L_{train}+0.3L_{misalign}$ | 73 | 66 | 34 | 72 |
> >
> > We observe that $\alpha=0.3$ produces results similar to $\alpha=0.1$. This strengthens our previous conclusions that the proposed alignment technique does not cause the model to disregard information from the front of the input sequence. We hope this additional analysis addresses your concern.
> >
> > ---
> >
> > Q2. Regarding the assumptions, I sincerely appreciate the extensive revision and the detailed explanations. As the experiments and qualitative analysis focus on vanilla Transformers, a brief discussion of how the assumption of linear attention could be extended or applied to vanilla Transformers would enhance the theoretical applicability and further strengthen the manuscript.
> >
> > A2. Thank you for your thoughtful suggestion. As discussed in [1], shallow linear Transformers share the same loss landscape as deep vanilla Transformers on tasks like linear regression and replicate key optimization features observed in full Transformers. This suggests that **linear Transformers serve as realistic abstractions for understanding Transformer optimization and generalization.**
> >
> > Building on this insight, our theoretical results derived under the linear attention assumption can provide meaningful guidance for vanilla Transformers. While linear attention simplifies the analysis, the fundamental principles of length generalization and output misalignment addressed in our framework are relevant to both models. The empirical success of our proposed methods, evaluated directly on vanilla Transformers in experiments like LongBench-E and BABILong, further supports this connection.
> >
> > We have added a brief discussion in Appendix B referencing [1] to clarify the relationship between the linear attention assumption and vanilla Transformers. We hope this addition addresses your concern.
> >
> > [1] Kwangjun Ahn et al, Linear attention is (maybe) all you need (to understand transformer optimization). ICLR, 2024.
> >
> > ---
> >
> > We hope these clarifications address your questions. Please let us know if there are any additional points we can address.

---

> > > ### Comment · Reviewer_AF82 · 2024-11-25
> > >
> > > Thank you for addressing my concerns.

---

### Official Review · Reviewer_1PA5 · 2024-11-03

**Soundness:** 2
**Presentation:** 3
**Contribution:** 2
**Rating:** 5
**Confidence:** 3

**Summary:**

This paper presents a novel approach to enhancing length generalization in large language models by focusing on output alignment (i.e., the consistency of output distributions across sequences of varying lengths), a departure from the more commonly explored input-focused strategies. The authors introduce a metric termed Long-Short Misalignment, which is shown to have a  correlation with length generalization performance. Additionally, the paper proposes a regularization loss aimed at improving this alignment and a reparameterization technique (OutRep) to explicitly align output distributions across different sequence lengths. Experimental results are provided to show that these techniques improve length generalization capabilities in both synthetic and natural language tasks.

**Strengths:**

1. Novel Perspective: The focus on output alignment as a means to improve length generalization is a fresh perspective that diverges from traditional input or positional encoding-based methods. This unique angle provides a valuable contribution to the ongoing discourse on enhancing transformer model capabilities.

2. Clear Presentation: The paper is well-written, with a clear presentation of ideas and experimental results.

**Weaknesses:**

1. Theoretical Issues: The theoretical proof of Theorem 3 could be problematic given the problematic definition of length generalization loss in the appendix (see detailed questions below), which weakens the theoretical soundness of the proposed approach. The empirical analysis cannot fully support the claim that minimizing the Long-Short Misalignment metric could lead to better length generalization  (see detailed questions below).

2. Computational Complexity: The implementation of the Long-Short Misalignment metric as a regularization loss introduces additional computational overhead. Moreover, the implementation details and complexity analysis, including their impact on model performance, are not adequately provided, which limits understanding of its scalability.

3. Lack of Comparison with Other Length Generalization Approaches: The paper does not provide a direct comparison between the proposed output alignment approach and other length generalization approaches (e.g., positional encoding-based approaches). Such a comparison would be beneficial to understand the relative strengths and weaknesses of the proposed method compared to existing approaches.

**Questions:**

1. Encouraging LLMs to be less sensitive to small input sequence length variations, given that the semantic meanings of these input sequences are very similar, is essentially a form of robustness. However, when input sequence length differences are significant, encouraging output alignment may not be appropriate, as the outputs are expected to differ substantially. In your regularization term computation, when you sample l1 and l2 from the interval [ltrain / 2, ltrain], could you conduct an ablation study examining how different sampling ranges |l1 - l2| affect performance?

2. The mathematical equation for length generalization loss (Eq. 43) defined in Theorem 3 in the appendix is problematic. It doesn't explicitly compare model performance across different input lengths. A more reasonable definition you might consider is Egen_{long inputs} / Egen_{short inputs} - 1, where Egen is your original definition (Eq. 43). Could you incorporate this definition into your analysis and discuss how it might affect the theoretical results and empirical findings?

3. In Tables 2 and 3, it might not make sense to compare the proposed approach with the baseline approach by setting the same training steps since the proposed approach requires more computation. Could you provide a comparison based on total computation time or FLOPs rather than training steps, which would give a more accurate picture of the efficiency-performance tradeoff?

4. Table 1 shows that more powerful LLMs, which achieve better evaluation scores on some long-context benchmarks, also tend to have smaller Long-Short Misalignment scores. However, it is insufficient to conclude that a smaller Long-Short Misalignment metric directly leads to better length generalization ability based solely on these results. Could you conduct additional experiments or analyses to establish a more direct link?

5. Can you compute the length generalization loss and add it to Tables 2 and 3 to better demonstrate that the proposed approach leads to improved length generalization?

6. There is a lack of details on the sampling strategies and complexity analysis of the Long-Short Misalignment regularization term computation, including their impact on model performance. Could you provide more information on these aspects?

7. How does the proposed output alignment technique compare with other length generalization approaches (e.g., positional encoding-based approaches)?

---

> ### Author Response · Authors · 2024-11-23
> **Reply to Reviewer 1PA5 (1/4)**
>
> We thank Reviewer 1PA5 for careful reading, detailed comments, as well as the appreciation of the novelty and clear presentation of our work. Below, we will address your concerns in the following points.
>
> ---
>
> Q1. When input sequence length differences are significant, encouraging output alignment may not be appropriate, as the outputs are expected to differ substantially. In your regularization term computation, when you sample $l_1$ and $l_2$ from the interval $[l_{train} / 2, l_{train}]$, could you conduct an ablation study examining how different sampling ranges $|l_1 - l_2|$ affect performance?
>
> A1. Sure. We consider four settings on the sampling ranges **$|l_1 - l_2|$**: (1) The current sampling strategy as described in the paper (2) Limiting $|l_1 - l_2|$ larger than $l_{train}/4$  (3) Limiting $|l_1 - l_2|$ smaller than $l_{train}/4$. (4) Uniformly sampling $l_1$ and $l_2$ from $[0, l_{train}]$. We conduct experiments using the same setting as Table 3 in the paper, using LongQLora for model adjustments and a regularization coefficient of $0.1$. The results are shown in the following table:
>
> | Benchmark |  | LongBench-E |  |  | Perplexity |  |
> | --- | --- | --- | --- | --- | --- | --- |
> | Training steps | 50 | 100 | 200 | 50 | 100 | 200 |
> | (1) Current sampling strategy | 21.8 | 23.3 | 25.8 | 6.72 | 6.39 | 5.77 |
> | (2) Limiting $\|l_1 - l_2\|\geq l_{train}/4$ | 21.5 | 23.2 | 25.7 | 6.77 | 6.29 | 5.81 |
> | (3) Limiting $\|l_1 - l_2\|\leq l_{train}/4$ | 21.4 | 22.7 | 24.5 | 6.82 | 6.47 | 5.94 |
> | (4) Sampling $l_1$ and $l_2$ from $[0, l_{train}]$ | 18.2 | 18.9 | 19.1 | 15.65 | 13.59 | 12.52 |
>
> The results show that: (1) **Setting 2** achieved performance comparable to the current strategy, while **Setting 3** showed slightly inferior performance compared to the current strategy. This suggests that aligning outputs between sequences with moderate length discrepancies effectively supports long-context modeling. (2) **Setting 4** yielded significantly worse performance than the current strategy, indicating that encouraging alignment between sequences with large length differences adversely affects the model's long-context capabilities.
>
> These results underscore the importance of carefully balancing the sampling range to optimize the model's generalization to longer contexts. We appreciate your suggestion and add this ablation study to Section 5 in the revision.
>
> ---
>
> Q2. The mathematical equation for length generalization loss (Eq. 43) defined in Theorem 3 in the appendix doesn't explicitly compare model performance across different input lengths.
>
> A2. We have revised the theoretical components of our work by adjusting certain assumptions within the overall framework. Specifically, we now make a moderate assumption that the model incorporates a linear attention block, which is also adopted by many previous theoretical analysis [1,2]. Under these revised assumptions, we achieve similar theoretical results:
>
> $$\epsilon(g_{\theta}^{l_{train}};l_{test})\leq L^2/l^2_{test}\cdot L_{misalign}(g_{\theta}^{l_{train}})+(L^2/l^2_{test}+1)\cdot L_{train}(g_{\theta}^{l_{train}})+C,$$
>
> where:
>
> - $\epsilon$ is the generalization loss of model $g_{\theta}^{l_{train}}$ with train length $l_{train}$,
> - $l_{test}$ is the test length,
> - $L=l_{test}-l_{train}/2$,
> - $L_{misalign}$ is the long-short misalignment loss,
> - $L_{train}$ is the training loss, and
> - $C$ is a constant
>
> The key difference from our prior work is that the coefficients of the two losses now depend on the training length $l$ and the test length $k$. This generalization guarantee provides two key insights:
>
> - **Role of Misalignment Loss**: Optimizing $L_{misalign}$ is critical for improving length generalization.
> - **Impact of Test Length**: As the test length $l_{test}$ increases, the coefficient of $L_{misalign}$ also grows, indicating that minimizing $L_{misalign}$ becomes increasingly important for long-context scenarios.
>
> We hope that this updated framework addresses the concerns.
>
> [1] Qi Zhang et al, Look ahead or look around? A theoretical comparison between autoregressive and masked pretraining. ICML, 2024.
>
> [2] Yue M. Lu et al, Asymptotic theory of in-context learning by linear attention. arXiv, 2405.11751.

---

> ### Author Response · Authors · 2024-11-23
> **Reply to Reviewer 1PA5 (2/4)**
>
> Q3. Could you provide a comparison based on total computation time or FLOPs rather than training steps in Table 2 and Table 3, which would give a more accurate picture of the efficiency-performance tradeoff?
>
> A3. Sure. We also admit the importance of evaluating the efficiency-performance tradeoff. Before presenting the comparison, we note that our method incurs only a marginal increase in computation cost—approximately 5% more than the baseline—due to the additional SCE loss introduced during training.
>
> Nevertheless, as you suggest, we conducted a comparison based on total computation time. Specifically, for equivalent computation costs: When the baseline is trained for 50, 100, and 200 steps, our method is trained for 47, 95, and 190 steps, respectively. The results of this comparison are shown in Table 9 and Table 10 in Appendix E and below.
>
> - Using the same setting as in Table 2. The first three rows of results use RedPajama-Book for training and the last three rows of results use PG19 for training. $(a/b)$ in the training steps mean $a$ steps for the baseline and $b$ steps for our method.
>
> | Benchmark |  | LongBench-E |  |  | Perplexity |  |
> | --- | --- | --- | --- | --- | --- | --- |
> | Training steps | 50/47 | 100/95 | 200/190 | 50/47 | 100/95 | 200/190 |
> | $L_{train}$(Baseline) | 22.7 | 23.8 | 24.7 | 7.21 | 6.56 | 6.12 |
> | $L_{train}+0.1L_{misalign}$ (Ours) | **23.1** | **25.1** | **26.4** | **6.92** | **6.27** | **5.91** |
> | $L_{train}+0.5L_{misalign}$ (Ours) | 21.7 | 23.4 | 24.5 | 7.62 | 7.16 | 6.61 |
> | $L_{train}$(Baseline) | 20.2 | 21.4 | 22.5 | **8.92** | **7.89** | 7.45 |
> | $L_{train}+0.1L_{misalign}$ (Ours) | **20.7** | **22.0** | **25.1** | 9.02 | 8.01 | **7.39** |
> | $L_{train}+0.5L_{misalign}$ (Ours) | 19.6 | **22.0** | 23.3 | 9.82 | 8.62 | 8.29 |
> - Using the same setting as Table 3. The first three rows of results use LongQLora for model adjustment methods and the last three rows of results use EABF for model adjustment methods. $(a/b)$ in the training steps mean $a$ steps for the baseline and $b$ steps for our method.
>
> | Benchmark |  | LongBench-E |  |  | Perplexity |  |
> | --- | --- | --- | --- | --- | --- | --- |
> | Training steps | 50/47 | 100/95 | 200/190 | 50/47 | 100/95 | 200/190 |
> | $L_{train}$(Baseline) | **21.9** | 22.1 | 23.4 | 6.82 | **6.41** | 5.82 |
> | $L_{train}+0.1L_{misalign}$ (Ours) | 21.6 | 23.1 | **25.7** | **6.79** | 6.42 | **5.74** |
> | $L_{train}+0.5L_{misalign}$ (Ours) | 21.1 | **23.6** | 24.9 | 7.19 | 6.65 | 5.96 |
> | $L_{train}$(Baseline) | 22.1 | 22.9 | 23.6 | **6.89** | 6.52 | 6.01 |
> | $L_{train}+0.1L_{misalign}$ (Ours) | **23.0** | **23.6** | **24.5** | 7.01 | **6.48** | **5.86** |
> | $L_{train}+0.5L_{misalign}$ (Ours) | 22.2 | 23.1 | 23.8 | 7.32 | 6.88 | 6.42 |
>
> We observe that the performance trends remain consistent: our method continues to outperform the baseline under equivalent computation time. This underscores the efficiency of our approach despite the minor additional cost.
>
> ---
>
> Q4. It is insufficient to conclude that a smaller Long-Short Misalignment metric directly leads to better length generalization ability based solely on these results in Table 1. Could you conduct additional experiments or analyses to establish a more direct link?
>
> A4. To address your concern, we have taken several steps:
>
> - Additional experiments: We conducted additional experiments under settings similar to those in Table 1. The results are shown below and also added to Table 1.
>
> | Model | $L_{train} | $L_{misalign}$ | $\log$(PPL) | LongBench-E Score |
> | --- | --- | --- | --- | --- |
> | RandomPos | 2.2 | 3.3 | 8.2 | 9.2 |
> | CLEX | 1.7 | 2.4 | 1.8 | 32.7 |
>
> These additional results strengthen the observed correlation between $L_{misalign}$ and long-context performance, providing robust empirical support for our assertion. The correlation coefficients remain consistently high across these extended experiments, enhancing the reliability of $L_{misalign}$ as a predictor of length generalization ability.
>
> - Clarification of Correlation v.s. Causation: We have revised the manuscript to ensure clarity that our results suggest a strong correlation between $L_{misalign}$ and length generalization ability, but **do not assert direct causation without theoretical and empirical validation**. Indeed, it is only through the combination of the theoretical generalization guarantees presented in Theorem 1 (in the revised manuscript) and the experimental results in Section 5—demonstrating that optimizing $L_{misalign}$ improves length generalization—that we can conclude a smaller $L_{misalign}$ contributes to better length generalization ability.
>
> These additions and clarifications have been incorporated into the revised manuscript. We hope these changes address your concerns and provide a stronger support for the findings presented in the paper.

---

> ### Author Response · Authors · 2024-11-23
> **Reply to Reviewer 1PA5 (3/4)**
>
> Q5. Can you compute the length generalization loss and add it to Tables 2 and 3 to better demonstrate that the proposed approach leads to improved length generalization?
>
> A5. Thank you for pointing this out. To clarify, the length generalization loss is directly related to the perplexity reported in Tables 2 and 3. Perplexity is the exponential of the cross-entropy loss. Thus, the perplexity values over the 8k-validation set in these tables can be interpreted as a measure of the length generalization loss with length of 8k.
>
> As shown in these tables, all our proposed methods trained with $\alpha=0.1$ for 200 steps achieve lower perplexity (and therefore better length generalization loss) compared to the baseline. This demonstrates the effectiveness of our approach in improving the model's length generalization capabilities.
>
> ---
>
> Q6. Could you provide more information on the sampling strategies and complexity analysis of the Long-Short Misalignment regularization term computation, including their impact on model performance?
>
> A6.
>
> **Sampling Strategies**
>
> We begin with the sampling strategies. As outlined in Section 4.2 of the paper, computing the Long-Short Misalignment regularization term $L_{misalign}$ involves sampling an integer $l_{extra}$ from the range $[1, l_{train}/2]$. A sequence of length $l_{train} + l_{extra}$ is then sampled from the training dataset. From this sequence, the first $l_{train}$ tokens form the first input sequence, while the last $l_{train}$ tokens form the second input sequence. The $L_{misalign}$ term is calculated using the overlapping positions between these two sequences.
>
> To investigate the impact of the sampling strategy, we conducted ablation studies using four configurations for $l_{extra}$:
>
> 1. The current strategy, sampling from $[1, l_{train}/2]$.
> 2. A narrower range, $[1, l_{train}/4]$.
> 3. A shifted range, $[l_{train}/4, l_{train}/2]$.
> 4. A broader range, $[1, l_{train}]$.
>
> Experiments were performed in the same setup as Table 3, using LongQLora for adjustments and a regularization coefficient of 0.1. The results are summarized below:
>
> - **Setting 2** achieved performance comparable to the current strategy.
> - **Setting 3** showed slightly worse performance, suggesting that moderate length discrepancies are crucial for effective long-context modeling.
> - **Setting 4** performed significantly worse, indicating that aligning sequences with large length differences negatively impacts long-context capabilities.
>
> These results highlight the importance of sampling ranges that balance input sequence similarity with moderate discrepancies, optimizing generalization to longer contexts. This ablation study has been incorporated into Section 5 of the revised manuscript.
>
> | Benchmark |  | LongBench-E |  |  | Perplexity |  |
> | --- | --- | --- | --- | --- | --- | --- |
> | Training steps | 50 | 100 | 200 | 50 | 100 | 200 |
> | (1) Current sampling strategy | 21.8 | 23.3 | 25.8 | 6.72 | 6.39 | 5.77 |
> | (2) Sampling $l_{extra}$ from a smaller range $[1, l_{train}/4]$ | 21.5 | 23.2 | 25.7 | 6.77 | 6.29 | 5.81 |
> | (3) Sampling $l_{extra}$ from a smaller range $[l_{train}/4, l_{train}/2]$ | 21.4 | 22.7 | 24.5 | 6.82 | 6.47 | 5.94 |
> | (4) Sampling $l_{extra}$ from a larger range $[1, l_{train}]$ | 18.2 | 18.9 | 19.1 | 15.65 | 13.59 | 12.52 |
>
> **Complexity Analysis**
>
> The computation of $L_{misalign}$ is based on output features. Given a vocabulary size of $|\mathcal{V}|$ and a training sequence length of $l_{train}$, the computation involves evaluating the SCE loss over $l_{train}/2 - l_{extra}$ positions. This entails:
>
> 1. Two logarithmic operations on matrices of shape $[l_{train}/2 - l_{extra}, |\mathcal{V}|]$.
> 2. Two point-wise multiplications on matrices of the same shape.
>
> As detailed in Equation (1) of the paper, these operations incur a computational cost that is significantly lower than the forward and backward passes of the model. Empirically, our approach increases computational time per step by only 3-5% relative to the baseline. Within equivalent training budgets, our method achieves superior performance, as also illustrated in Answer 3 and Table 9 and Table 10 in Appendix E.

---

> ### Author Response · Authors · 2024-11-23
> **Reply to Reviewer 1PA5 (4/4)**
>
> Q7. How does the proposed output alignment technique compare with other length generalization approaches (e.g., positional encoding-based approaches)?
>
> A7. We address the reviewer's question by conducting experiments under the following settings:
>
> 1. **Raw fine-tuning**: The baseline without any additional techniques.
> 2. **Only LongQLora**: Incorporating the LongQLora method as described in Table 3. LongQLora leverages multiple techniques to extrapolate, including Position Interpolation, QLoRA, and Shift Short Attention from LongLoRA.
> 3. **Only EABF**: Incorporating the EABF method as described in Table 3. EABF introduces a dynamic rescaling mechanism to the attention layers and applies a higher base frequency for RoPE to extrapolate.
> 4. **Only our proposed output alignment technique**: Applying our method without other enhancements.
>
> The results are summarized in the table below and in Table 11 in the revision.
>
> | Benchmark |  | LongBench-E |  |  | Perplexity |  |
> | --- | --- | --- | --- | --- | --- | --- |
> | Training steps | 50 | 100 | 200 | 50 | 100 | 200 |
> | (1) Raw finetuning | 14.9 | 16.1 | 18.4 | 21.73 | 18.52 | 16.13 |
> | (2) Only with LongQLora | 21.9 | 22.1 | 23.4 | 6.82 | 6.41 | 5.82 |
> | (3) Only with EABF | 21.4 | 22.7 | 24.5 | 6.89 | 6.52 | 6.01 |
> | (4) Only with output alignment technique ($\alpha=0.1$) | 15.4 | 18.2 | 20.9 | 19.25 | 15.47 | 11.24 |
>
> From the results, we observe:
>
> - Our method alone outperforms raw fine-tuning, demonstrating its effectiveness in improving length generalization.
> - Positional encoding-based approaches, such as LongQLora and EABF, achieve higher performance compared to using our method alone.
>
> When combined with positional encoding-based approaches, our method consistently yields additional improvements, as shown in Tables 2 and 3 in the paper. It is important to note that our proposed output alignment technique operates from the perspective of loss design, focusing on the model's output alignment during training. In contrast, positional encoding-based approaches primarily address the input representation. **These two strategies are orthogonal and can be seamlessly integrated.** The experimental results demonstrate that incorporating our method with positional encoding techniques enhances long-context modeling capabilities beyond what is achieved by either approach alone.
>
> ---
>
> Thanks for your careful reading and detailed review. Hope our explanations and extended empirical justifications could address your concern. Please let us know if you have additional questions.

---

> ### Author Response · Authors · 2024-11-25
> **Your invaluable input is needed.**
>
> Dear Reviewer 1PA5,
>
> Thank you for your thoughtful feedback on our manuscript. We have carefully addressed each of your questions in our detailed response. We would greatly appreciate it if you could review our revisions and let us know if they address your concerns.
>
> We are encouraged by Reviewer AF82’s acknowledgment of our revisions, which they found satisfactory, and their decision to raise the score from 3 to 6. Your further input would be invaluable to us as we continue to refine our work.
>
> Thank you again for your time and effort, and we hope you have a wonderful day!
>
> Best regards,
>
> Authors

---

> ### Comment · Reviewer_1PA5 · 2024-11-26
>
> Thank the authors for the detailed response which addressed some of my concerns. I have updated my rating accordingly.

---

> > ### Author Response · Authors · 2024-11-26
> > **Are there any remaining concerns?**
> >
> > Dear Reviewer 1PA5,
> >
> > Thank you for your thoughtful feedback and for raising your score. We appreciate your acknowledgment that our response addressed some of your concerns.
> >
> > We noticed that your updated rating remains at 5, marginally below the acceptance threshold. If there are any remaining concerns or aspects of our work that you feel require further clarification or improvement, we would be grateful if you could share them with us. We are committed to addressing your feedback to the best of our ability.
> >
> > Thank you again for your time and effort in reviewing our manuscript. We greatly value your insights and hope you have a wonderful day!
> >
> > Best regards,
> >
> > Authors

---

> ### Author Response · Authors · 2024-12-02
> **Your further inputs are greatly appreciated. Only one day left.**
>
> Dear Reviewer 1PA5,
>
> Thank you again for your thoughtful feedback and for raising your score. We noticed, however, that your updated rating of 5 remains “marginally below the acceptance threshold.”
>
> With the discussion period ending in about one day, we kindly ask if there are any remaining concerns we could address to further clarify or improve our work. Your insights are invaluable, and we are eager to respond.
>
> Thank you for your time and consideration. Wishing you a great day!
>
> Best regards,
> Authors

---

### Official Review · Reviewer_v4pP · 2024-11-04

**Soundness:** 2
**Presentation:** 2
**Contribution:** 2
**Rating:** 6
**Confidence:** 4

**Summary:**

This paper proposes a new view of achieving length generalization of LLMs by the output distribution adjustment/alignment. It identifies output alignment, the consistency of output distributions across varying sequence lengths, as a key factor in length generalization. The authors introduce the Long-Short Misalignment metric to quantify this alignment and demonstrate its strong correlation with models' performance on long-context tasks. Hence, they propose a regularization loss based on this metric, which, when incorporated into training, significantly improves models' ability to handle longer sequences. The evaluation is conducted on language modeling (Redpajama and PG19) and LongBench, where the proposed method demonstrates certain effectiveness.

**Strengths:**

1. Considering that inputs of variable lengths significantly vary the output of LLMs, the view of achieving length generalization by addressing the output distribution gap between short and long inputs is novel and makes sense.

2. The evaluation of language modeling and LongBench demonstrated the effectiveness of the proposed misalign loss and verified the authors' postulation.

**Weaknesses:**

1. In your settings, the short input is usually the truncated local context window of a long input, and the output distribution from the short input would serve as the "ground truth" distribution in SCE loss in Eq. (5). However, if the output relies on previous context information that's been removed, the output distribution given short input is also divergent and even wrong. For such a situation, the alignment may mislead the long-context output distribution and limit the capabilities (the model may tend to attend to local context only).

2. The models in experiments are trained based on CLEX. CLEX and some similar methods (e.g., YaRN) would dynamically adjust the frequency of RoPE according to the input length. In your implementation (line #376 to #404), the long and short inputs would be incorporated into one sentence, which would make the "short output distribution" be accessed based on "long RoPE frequency". Such short output distribution may deviate from the naive one and potentially affect the model's capabilities.

3. The evaluation length is up to 8K, which may be insufficient for a method designed for length generalization. I believe the evaluations over longer sequences may help to validate the effectiveness of the proposed method.

4. Section 3 seems to be redundant. The motivation and the proposed method are straightforward but the case study in section 3 is somewhat difficult to follow.

**Questions:**

Please refer to the weaknesses.

---

> ### Author Response · Authors · 2024-11-23
> **Reply to Reviewer v4pP (1/3)**
>
> We thank Reviewer v4pP for careful reading and detailed comments of our work. Below, we will address your concerns about the long-short alignment, the evaluation method as well as the clarity of Section 3.
>
> ---
>
> Q1. If the output relies on previous context information that's been removed, the output distribution given short input is divergent. For such a situation, the alignment may mislead the long-context output distribution and limit the capabilities (the model may tend to attend to local context only).
>
> A1. Indeed, encouraging alignment in such cases may introduce biases that obstruct the model's ability to utilize distant context. To address this, we have constrained the length sampling interval for the two sequences to $[l_{train}/2, l_{train}]$, ensuring that alignment is only encouraged between sequences with moderate length discrepancies. This design choice avoids scenarios where excessive divergence in input lengths could mislead the output alignment process.
>
> To ensure that the model will not tend to attend to local contexts only, we conduct experiments on the BABILong benchmark [1], a challenging reasoning-in-a-haystack task designed for long-context evaluation. BABILong consists of question-answering tasks where the supporting fact for each question is located at a specific position within the context. We analyze the impact of the supporting fact's position within the input context using the QA1 task from BABILong, where each question is associated with a single supporting fact. Results from this analysis are presented in Table 7 in the revision and below:
>
> |  | Fact Depth=0% | Fact Depth=25% | Fact Depth=50% | Fact Depth=75% |
> | --- | --- | --- | --- | --- |
> | $L_{train}$ (Baseline) | 75 | 64 | 30 | 69 |
> | $L_{train}+0.1L_{misalign}$ | 73 | 64 | 38 | 74 |
>
> When the supporting fact is located at the beginning of the input context (fact depth = 0), our method achieves performance comparable to the baseline (73 v.s. 75). This finding suggests that the proposed alignment technique does not restrict the model's ability to attend to distant positions in the context, even when the supporting fact is located far from the query position.
>
> [1] Yuri Kuratov et al, Babilong: Testing the limits of llms with long context reasoning-in-a-haystack, arxiv 2406.10149.
>
> ---
>
> Q2. CLEX would dynamically adjust the frequency of RoPE according to the input length. In your implementation, the "short output distribution" is accessed based on "long RoPE frequency”. Such short output distribution may deviate from the naive one and potentially affect the model's capabilities.
>
> A2. To address your concern, we note that during fine-tuning, the baseline method, CLEX, employs next-token prediction for the entire input sequence. This process inherently involves using the long RoPE frequency, even when generating predictions conditioned on short prefixes of the input sequence.
>
> In this context, the computation of the short output distribution is also based on the long RoPE frequency, as it arises from predictions over chunks of the full input sequence rather than isolated short sequences. Our approach follows the same principle, ensuring consistency with the behavior of CLEX during fine-tuning.
>
> Therefore, the method we employ to calculate the short output distribution aligns with the baseline's implementation and maintains validity within this framework.

---

> ### Author Response · Authors · 2024-11-23
> **Reply to Reviewer v4pP (2/3)**
>
> Q3. The evaluation length is up to 8K, which may be insufficient for a method designed for length generalization. I believe the evaluations over longer sequences may help to validate the effectiveness of the proposed method.
>
> A3. There may be some misunderstanding about the evaluation. In fact, the evaluation length in LongBench-E that we use in the paper reaches up to 32K, which is considered an effective evaluation benchmark for length generalization [2, 3]. We have added this explanation in the setting part of Section 5.1 in the revision. Additionally, we conduct evaluation experiments on BABILong [1], where the input length can reach 16K. The results can be viewed in Table 6 and Table 7. We also copy these results in the following tables:
>
> - Evaluation is performed on input sequences of lengths 4K, 8K, and 16K, with the overall results summarized in Table 6.
>
> | Evaluation Length | 4K | 8K | 16K |
> | --- | --- | --- | --- |
> | $L_{train}$ (Baseline) | 48.2 | 42.4 | 37.9 |
> | $L_{traiin}+0.1L_{misalign}$ | **49.1** | **44.4** | **40.1** |
>
> The results indicate that our proposed method consistently outperforms the baseline across all evaluated lengths. Specifically, our method achieves a performance gain of $2.0\%$ at length 8K and $2.2\%$ at length 16K. These results demonstrate the effectiveness of our regularization loss in enhancing length generalization.
>
> - Additionally, we analyze the impact of the supporting fact's position within the input context using the QA1 task from BABILong, where each question is associated with a single supporting fact. Results from this analysis are presented in Table 7.
>
> |  | Fact Depth=0% | Fact Depth=25% | Fact Depth=50% | Fact Depth=75% |
> | --- | --- | --- | --- | --- |
> | $L_{train}$ (Baseline) | 75 | 64 | 30 | 69 |
> | $L_{train}+0.1L_{misalign}$ | 73 | 64 | 38 | 74 |
>
> These results offer two key insights: (1) **Performance with early-context facts:** When the supporting fact is located at the beginning of the input context (fact depth = 0), our method achieves performance comparable to the baseline. This suggests that, despite the form of the regularization potentially encouraging the model to neglect earlier contexts, it does not lead to this behavior in practice. (2) **Performance with middle-context facts:** When the supporting fact is positioned in the middle of the context (fact depth = 50 or 75), our method shows considerable improvement over the baseline. This indicates that our approach effectively mitigates the "loss-in-the-middle" phenomenon [4], a common challenge in large language models.
>
> The full setting and results can be viewed in Section 5.4 in the revision. We hope these results can ease your concern on the effectiveness of the proposed method.
>
> [1] Yuri Kuratov et al, Babilong: Testing the limits of llms with long context reasoning-in-a-haystack,
> arxiv 2406.10149.
>
> [2] Guanzheng Chen et al, Clex: Continuous length extrapolation for large language models. ICLR, 2024.
>
> [3] Hongye Jin et al, LLM maybe longlm: Self-extend LLM context window without tuning. ICML, 2024.
>
> [4] Nelson F Liu et al, Lost in the middle: How language models use long contexts. Transactions of the Association for Computational Linguistics, 2024.

---

> ### Author Response · Authors · 2024-11-23
> **Reply to Reviewer v4pP (3/3)**
>
> Q4. Section 3 seems to be redundant. The motivation and the proposed method are straightforward but the case study in section 3 is somewhat difficult to follow.
>
> A4. We follow your suggestion and refine Section 3 to slightly more than 1 page in the revision by removing the theoretical part and most of the explanation. Now, Section 3 only serves as a motivation for pointing out the important role of output space in the length generalization, as the empirical results in the synthetic tasks are direct and significant.  Subsequently, we add more explanation and empirical results in Section 4 and Section 5, making the main focus of the paper to be the phenomenon and solution in the natural language task. We also add Section 6 for discussion about the relation between synthetic tasks and natural language tasks. We hope that this revision can improve the clarity and fluency of the paper.
>
> ---
>
> Thanks again for your detailed comments, which are very helpful, and hope our response could address your concerns. Please let us know if you have additional questions.

---

> ### Author Response · Authors · 2024-11-25
> **Your invaluable input is needed.**
>
> Dear Reviewer v4pP,
>
> Thank you for your thoughtful feedback on our manuscript. We have carefully addressed each of your questions in our detailed response. We would greatly appreciate it if you could review our revisions and let us know if they address your concerns.
>
> We are encouraged by Reviewer AF82’s acknowledgment of our revisions, which they found satisfactory, and their decision to raise the score from 3 to 6. Your further input would be invaluable to us as we continue to refine our work.
>
> Thank you again for your time and effort, and we hope you have a wonderful day!
>
> Best regards,
>
> Authors

---

> ### Author Response · Authors · 2024-12-02
> **Your further inputs are greatly appreciated. Only one day left.**
>
> Dear Reviewer v4pP,
>
> For your raised questions, we prepared a detailed response to address your concerns. We were hoping to hear your feedback on them.
>
> As there is only one day left for the reviewer and author discussions and we understand that everyone has a tight schedule, we kindly wanted to send a gentle reminder to ensure that our response sufficiently addressed your concerns or if there are further aspects we need to clarify.
>
> If you could find the time to provide your thoughts on our response, we would greatly appreciate it.
>
> Best, Authors

---

> > ### Comment · Reviewer_v4pP · 2024-12-03
> >
> > Really appreciate the authors' thorough responses to the raised questions. My concerns regarding local context truncation,  length generalization, and paper writing have been largely resolved and I decided to increase my rating to 6.

---

> > > ### Author Response · Authors · 2024-12-04
> > > **Thank You for Your Feedback!**
> > >
> > > Dear Reviewer v4pP,
> > >
> > > Thank you for your thoughtful feedback and for increasing your rating. We’re glad our responses addressed your concerns. We greatly appreciate your time and insights, which have been invaluable in improving our work.
> > >
> > > Best regards,
> > >
> > > Authors

---

### Author Response · Authors · 2024-11-23
**Updates in the manuscript**

We sincerely thank all reviewers for their efforts, detailed comments, and thoughtful suggestions. We are encouraged by all the reviewers’ recognition of the novelty of our work. In response to their feedback, we have updated the manuscript, with the main changes highlighted in blue. These updates include:

1. **Revision of Section 3:**
We refined Section 3 to more clearly emphasize the motivations behind our work. This section has been condensed to slightly more than one page, with detailed theoretical content (Original Theorem 1 and Original Theorem 2) and explanations moved to the appendix. This refinement aims to make the purpose of the synthetic tasks clearer.

2. **Theoretical Adjustments:**
We revised the theoretical framework in Appendix B by adjusting certain assumptions to strengthen the validity of our conclusions.

3. **Discussion of Synthetic vs. Natural Language Tasks:**
Section 6 now includes an in-depth discussion of the relationship and comparisons between synthetic tasks and natural language tasks.

4. **Additional Experiments:**
We conducted several new experiments to address the reviewers’ suggestions:
- **Ablation Studies:** We performed ablation studies on the regularization coefficient $\alpha$ and the sampling range (Section 5.3), providing guidance for hyperparameter selection.
- **Evaluation on BABILong Benchmark:** Results in Section 5.4 demonstrate the effectiveness of our method in improving length generalization.
- **Comparison under Equivalent Computation Time:** Appendix E includes results showing our method consistently outperforms the baseline when compared under the same computational cost.
- **Additional Dataset Experiments:** Appendix F supplements the original mean prediction task experiments with a different dataset setting.
- **Comparison Without Model Adjustment:** Appendix G presents experiments to compare our approach without applying model adjustment methods.

---

### Meta-Review · Area_Chair_ynD7 · 2024-12-26

**Metareview:**

Reviewers find the papers approach novel and interesting at addressing length generalization for LMs. However they raised significant concerns about the initial manuscript - concerns about assumptions and proof of the theorems, limited evaluations for long context generalization. In response authors significantly revised manuscript introducing new assumptions and updating the theoretical results. they also present more long context evaluations. While reviewers appreciate the new experimental results, they do not find sufficient contributions and hence do not recommend acceptance at this stage. Overall I think the paper is on borderline, the paper can use another round of rewrite with the substantial changes to the theoretical analysis and the new experimental results, and can benefit from another round of reviews. Hence I recommend rejection.

**Additional Comments On Reviewer Discussion:**

Reviewers raised concerns about the assumptions and the theoretical analysis, limited long context evaluations. Authors updated the assumptions and revised the analysis. They also presented additional long context evaluations.

---

### Decision · Program_Chairs · 2025-01-22

Reject